# Current Understanding of the Structure and Function of Pentapeptide Repeat Proteins

**DOI:** 10.3390/biom11050638

**Published:** 2021-04-26

**Authors:** Ruojing Zhang, Michael A. Kennedy

**Affiliations:** Department of Chemistry and Biochemistry, 106 Hughes Laboratories, Miami University, Oxford, OH 45056, USA; zhangr18@miamioh.edu

**Keywords:** cyanobacteria, filamentous cyanobacteria, heterocyst, *Nostoc* sp. strain Pcc7120, pentapeptide repeat protein, protein crystallography, repeat five residue fold

## Abstract

The pentapeptide repeat protein (PRP) superfamily, identified in 1998, has grown to nearly 39,000 sequences from over 3300 species. PRPs, recognized as having at least eight contiguous pentapeptide repeats (PRs) of a consensus pentapeptide sequence, adopt a remarkable structure, namely, a right-handed quadrilateral β-helix with four consecutive PRs forming a single β-helix coil. Adjacent coils join together to form a β-helix “tower” stabilized by β-ladders on the tower faces and type I, type II, or type IV β-turns facilitating an approximately −90° redirection of the polypeptide chain joining one coil face to the next. PRPs have been found in all branches of life, but they are predominantly found in cyanobacteria. Cyanobacteria have existed on earth for more than two billion years and are thought to be responsible for oxygenation of the earth’s atmosphere. Filamentous cyanobacteria such as *Nostoc* sp. strain PCC 7120 may also represent the oldest and simplest multicellular organisms known to undergo cell differentiation on earth. Knowledge of the biochemical function of these PRPs is essential to understanding how ancient cyanobacteria achieved functions critical to early development of life on earth. PRPs are predicted to exist in all cyanobacteria compartments including thylakoid and cell-wall membranes, cytoplasm, and thylakoid periplasmic space. Despite their intriguing structure and importance to understanding ancient cyanobacteria, the biochemical functions of PRPs in cyanobacteria remain almost completely unknown. The precise biochemical function of only a handful of PRPs is currently known from any organisms, and three-dimensional structures of only sixteen PRPs or PRP-containing multidomain proteins from any organism have been reported. In this review, the current knowledge of the structures and functions of PRPs is presented and discussed.

## 1. Introduction

The first description of a pentapeptide repeat protein (PRP) was reported in 1995, when Haselkorn and coworkers identified a gene from the filamentous cyanobacterium *Nostoc* (formerly *Anabaena)* sp. strain PCC 7120, which, when mutated, altered the composition of glycolipids encasing the heterocysts, among other alterations (Figure 1) [1]. They named the gene *heterocyst-specific glycolipids-directing protein K* (*hglK)* for the role that the gene played in localization of glycolipids to heterocysts. The protein encoded by the *hglK* gene, HglK, was predicted to contain four trans-membrane spanning regions and an unusual alanine- and leucine-rich pentapeptide repeat (PR) region made up of 36 PRs with the consensus sequence AXLXX [1]. Therefore, HglK was the first PRP to be associated with a putative biochemical function. To date, however, the precise mechanism for the role that HglK plays in regulating glycolipid localization to heterocysts remains unknown. 

In 1998, Bateman et al. [2] reported the discovery of a novel family of proteins, to which HglK belonged, that contained tandem PRs with the sequence motif A(D/N)LXX, based on the analysis of recently determined complete genomes of several bacteria at the time. They observed that PRPs were most commonly found in cyanobacteria [2]. The authors also proposed a model of PRP structures, rightly predicting that PRPs would adopt a right-handed β helical architecture; however, they predicted a triangular-shaped helix, which would prove to be in error once the first three-dimensional structures of PRPs were determined several years later. 

Furthermore, in 1998, Martínez-Martínez confirmed that quinoline resistance in bacteria could be carried on a multi-resistance plasmid (pMG252), which they discovered in a clinical isolate of *Klebsiella pneumonia* [3]. Bacterial acquisition of antimicrobial resistance undergoes constant evolution with horizontal gene transfer through plasmids playing a major role [4]. In 2001, Montero et al. discovered that intrinsic resistance to fluoroquinolines was influenced by MfpA, a putative PRP-containing protein encoded by a chromosomal gene in *Mycobacterium smegmatis* [5]. In 2005, Hegde et al. [6] solved the three-dimensional structure of MfpA from *Mycobacterium tuberculosis*, a homologue of MfpA from *M. smegmatis,* representing the first three-dimensional structure of a PRP, revealing that it adopted a right-handed quadrilateral β helical structure. Hegde et al. also reported that the structure and electrostatic charge distribution of MfpA mimicked that of DNA, and therefore MfpA was able to confer fluoroquinoline resistance to *M. tuberculosis* due to its ability to mimic DNA, bind to DNA gyrase and inhibit its function [6]. Soon after, many more chromosomal genes encoding homologs of MfpA were discovered in the genomes of a variety of organisms and in 2013, Jacoby and Hooper reported a phylogenetic tree analysis that showed that quinoline resistance genes (*qnr)* and *mfpA* homologs could be identified in the chromosomes of 58 Gram-negative bacteria and 34 Gram-positive organisms and in 14 plasmid-mediated genes [7]. 

In 2003, Chandler et al. [8] ascribed a cellular function to RfrA, a PRP from the cyanobacterium *Synechocystis* 6803, showing that it played a role in regulating a novel manganese uptake system; however, the nature of the system and the precise role that RfrA plays in regulating the manganese uptake system remains unknown.

By 2006, Vetting et al. [9] reported that the PRP family had grown to more than 500 members in the prokaryotic and eukaryotic kingdoms and they updated the PR consensus sequence as [S,T,A,V][D,N][L,F][S,T,R][G]. In 2009, Buchko reviewed the knowledge of the structure and function of PRPs from cyanobacteria [10]. In 2014, Shah and Heddle reported that a query of the Pfam database (http://pfam.xfam.org (accessed on 15 June 2014)) for members of the PR family (PF00805) had expanded to 11,082 sequences from 1513 species [11] and that protein structures had been solved for a number of PRPs from *Nostoc.* sp. PCC 7120 [12,13], Cyanothece 51142 [14,15], *Arabidopsis thaliana* [16], *Enterococcus faecalis* [17], *K. pneumonia* [18], *Xanthomonas albilineans* [19], *Aeromonas hydrophila* [20], and *M. tuberculosis* [6]. In 2019, Zhang et al. updated the PR consensus sequence to (A/C/S/V/T/L/I)/(D/N/S/K/E/I/R)/(L/F)/(S/T/R/E/Q/K/V/D)/(G/D/E/N/R/Q/K) based on the consideration of several newly available PRP crystal structures [21]. By 2020, the number of PRPs in the PF00805 Pfam had increased to 38,000 sequences in over 3300 species reported in a study by Xu and Kennedy that characterized the protein dynamics in PRPs [22]. The current distribution of PRPs in PF00805 Pfam is depicted in Figure 2, indicating 38,981 PRP sequences distributed over 3338 species (https://pfam.xfam.org/family/PF00805#tabview=tab7 (accessed on 25 April 2021)). In this sunburst plot, 82.2% of the species and 84.7% of the sequences belong to bacteria, 14.1% of species and 13.7% of sequences belong to eukaryota, 0.5% of species and 1.4% of sequences belong to viruses, and 1.1% of species and 2.2% of sequences belong to archaea. The plot shows that PRPs are found most abundantly in cyanobacteria, with 26.9% of all PRP sequences occurring in cyanobacteria; however, cyanobacteria represent only 3.7% of the species in which PRPs have been discovered, indicating that PRPs likely played an important physiological or structural role in the evolution and lifecycle of ancient cyanobacteria. Despite the large and growing nature of the PRP superfamily, three-dimensional structures of only sixteen PRP or PRP-containing proteins have been determined, thirteen of which contain a single PRP domain with α helices capping the N and/or C termini and three of which contain two or more domains including the PRP domain. 

In this review, several PRP categories are discussed, ranging from those that have putative associated biochemical or cellular functions to those that have had structures determined but with unknown putative functions, including those involved in (1) heterocyst glycolipid synthesis, (2) manganese uptake, (3) gyrase inhibition, (4) ubiquitin E3 ligases, (5) synaptic vesicle glycoprotein 2 isoform C (SV2C) receptors, and (6) plant and cyanobacteria proteins with three-dimensional structures but no functional characterization (Figure 3). Although the biological functions of most PRPs remain unknown, three-dimensional structures of PRPs and PRP-containing multidomain proteins continue to be solved and reported with the hope of helping to eventually understand their biological, biochemical or cellular functions.

## 2. Cyanobacterial and Eubacterial PRPs with an Associated Biochemical or Cellular Function

### 2.1. Heterocyst Glycolipid Biosynthesis-HglK

In 1995, Haselkorn and coworkers identified the *hglK* gene in mutant strain 543 of the filamentous *Nostoc* sp. strain PCC 7120 [1]. The mutant strain was isolated as a $Fox^{-}$ (lack of ability to fix dinitrogen except in an oxygen-depleted environment) mutant [23] following chemical mutagenesis [24]. The ultrastructural phenotype of the mutant strain showed that, in nitrogen replete media, i.e., media containing an abundant usable soluble nitrogen source, the vegetative cells in the filaments, i.e., those cells capable of dividing and extending the filament length, were more cylindrical and had thicker septa compared to the wild-type strain, whereas in nitrogen depleted media, the mutant heterocysts lacked the glycolipid layer that is normally exterior to the cell wall and isolates the nitrogenase enzyme that is required for fixation of atmospheric nitrogen inside the heterocysts from oxygen that inactivates the nitrogenase enzyme. Hydropathy analysis indicated that the 727 amino acid HglK protein contained four potential trans-membrane regions in its N-terminal region and 36 PRs in its C-terminal region starting at amino acid position 501. Analysis of the mutant strain indicated that it contained a stop codon just upstream of the DNA encoding the PRP domain. Because heterocysts in the mutant strain lacked the glycolipid layer exterior to the cell wall, the authors used thin-layer chromatography to analyze the lipid content of the mutant and wild-type strains and found no difference. The authors therefore concluded that the *hglK* gene encoded a protein that was necessary for localization of glycolipids to the heterocyst walls and that this function required the PRP domains.

Ar *é* valo and Flores further characterized the function of the *hglK* gene and discovered that *hglK* mutants were also defective in heterocyst differentiation, being impaired in the expression of the heterocyst-related genes *coxB2A2C2* (a cytochrome c oxidase) and *nifHDK* (a nitrogenase) [25]. The authors also observed that HglK was predominantly localized at the intercellular septa and was required for biogenesis of long filaments, to produce normal numbers of nanopores, and for normal intercellular molecular transfer activity. The authors concluded that HglK contributed to the architecture of the intercellular septa and impacted the function of septal junctions [25]. The precise biochemical role of HglK remains unknown and no three-dimensional structure of HglK is currently available.

### 2.2. Regulator of Manganese Uptake-RfrA

In 2003, Pakrasi and coworkers discovered that RfrA (gene name *sll1350*), a PRP from *Synechocystis* 6803, was a regulator of a novel high-affinity manganese uptake system [8]. RfrA and its function were identified in a suppressor screen in which the mutant strain was deficient in both *mntC*, a gene encoding a component of an ABC transport system for manganese, and *psbO*, which encoded an extrinsic manganese stabilizing protein of photosystem II. In a suppressor screen, one looks for additional mutations that reverse the mutant phenotype, in this case, deficiency in manganese transport. The authors discovered that a point mutation in rfrA restored photosynthetic activity of the *Δ**mntC*, *Δ**psbO* double deletion mutant. Radioactive manganese uptake experiments indicated that RfrA was a regulator of a high affinity manganese transport system that was different from the known manganese ABC transport system. The authors named the 398 amino acid RfrA protein for the repeat five-residues (Rfr) domain, which is another name for PRPs, which occurred in the N-terminus of the protein. Genetic analysis indicated that *Synechocystis* 6803 contained sixteen PRPs. RfrA was the first member of the PRP family to be linked to a specific physiological process. RfrA has no sequence or structural similarities to previously described bacterial manganese transcription factors and it does not have any known DNA-binding domains [26]. It has been postulated that RfrA may regulate the second manganese transporter through a mechanism other than transcriptional control, such as by reversible protein modifications at the post-translational level [26]. Despite the link to regulation of manganese uptake, the nature of the hypothetical second high-affinity manganese importer, its regulatory mechanism, and the precise biochemical role that RfrA plays regulating the putative manganese transporter remains unknown [26]. The three-dimensional structure of RfrA also remains unknown.

### 2.3. Gyrase Inhibitors

#### 2.3.1. MfpA (2BM4, 2BM5, 2BM6 and 2BM7) 

In the early 2000s, with the growing antibiotic resistance of *M. tuberculosis* to two bactericidal compounds, isoniazid and rifampicin [27], fluoroquinolines had become the most common new antibiotic therapy to treat *M. tuberculosis* infections [6,27]. In 2001, Montero et al. identified a gene *mfpA* that encoded a PRP that conferred a new mechanism of fluoroquinoline resistance to *M. smegmatis* [5]. The protein encoded by the *mfpA* gene resulted in a low level of resistance to ciprofloxacin and sparfloxacine [5]. Hegde et al. [6] identified a 183-amino acid MfpA homolog from *M. tuberculosis* (MtMfpA) encoded by the Rv3361c gene that was 67% identical to the 192-residue *M. smegmatis* MfpA protein. In 2005, Hegde et al. [6] reported the three-dimensional structure of MfpA from *M. tuberculosis* revealing it to be a PRP made up of eight complete PR coils composed of a mixture of type II and type IV β turns (Figure 4). MfpA was the first PRP to have its three-dimensional structure solved. Hegde et al. reported that MfpA expression in vivo conferred resistance to the antibiotic fluoroquinolone. Fluoroquinolines are chemotherapeutic bactericidal drugs that interfere with DNA replication in bacteria, leading to bacterial cell death. Fluoroquinolines exert their antibacterial activity by interfering with the normal function of the type II topoisomerases, DNA gyrase and DNA topoisomerase IV. These enzymes normally cut the genomic DNA to allow supercoiling and then ligate the DNA to stabilize the supercoiled DNA. Fluoroquinoline acts by inhibiting the ligase activity of these enzymes and leaving the nuclease activity intact, resulting in accumulation of single- and double-strand breaks that leads to disrupted DNA replication and cell death. Fluoroquinoline acts by binding reversibly to the gyrase-DNA complexes and stabilizing the covalent enzyme tyrosyl-DNA phosphate ester that is normally a transient intermediate in the topoisomerase reaction. Hegde et al. reported that the three-dimensional structure of MfpA exhibited a size, shape and electrostatic surface similar to that of B-form DNA, and concluded that its mechanism of action was due to DNA mimicry [6]. Since fluoroquinoline only binds to DNA gyrase-DNA complexes, binding of MfpA to DNA gyrase blocks fluoroquinoline binding to DNA gyrase-DNA complexes, thus interfering with the bactericidal activity of fluoroquinoline [6]. 

Because its function relies on a mechanism of DNA mimicry, MfpA exists as functional dimer. Two molecules of MfpA undergo a head-to-head interaction mediated by two α helices at the C terminus of each molecule to form an asymmetric rod-like shape with a hydrophobic dimer interface (Figure 4, top panel). Based on the electrostatic surface potential of the MfpA dimer, Hegde et al. generated a model in which the β helices of the MfpA dimer interacted with N terminal region of gyrase dimer through electrostatic interactions. The secondary structure of MfpA contains a β bulge in its β helix and two α helices at the C terminus and 28 to 29 β turns (Figure 4, top panel). 

The head-to-head interaction two molecules of MfpA involves 81 contacts mediated by 18 residues on one chain and 17 residues on the second chain that interact through hydrogen bonds and non-bonded contacts involving a surface area of about 1700 Å^2^ (Figure 4, bottom panel). An interweaved interaction between two MfpA molecules is mediated by a short parallel β sheet involving residues 162–164 on one strand (chain A) and residues 178–181 on the second strand (chain B), an orthogonal interaction between α-helices (165–176 on each MfpA molecule), and another short parallel β sheet involving residues 178–181 (chain A) and 162–164 (chain B) (Figure 4, bottom panel). The dimer interaction was stabilized by a hydrophobic core formed by the interacting hydrophobic side chains from each α-helix primarily involving F172 (12 interactions), H176 (2 interactions), L178 (9 interactions), plus the H-bonds stabilizing the parallel β-sheets, and multiple interactions involving R145 (six interactions), R164 (7 interactions), and C179 (7 interactions) (Figure 4, bottom panel). The energy to form the dimer interaction was −15.8 kcal/mole (2BM4). Considering, that a single hydrogen-bond has a typical energy of 1–3 kcal/mol, this indicates that the head-to-head dimer formation is energetically favored (Figure 4, bottom panel).

#### 2.3.2. EfsQnr (2W7Z)

In 2007, Arsene and Leclercq discovered a qnr-like gene from *E. faecalis*, EfsQnr, which conferred fluoroquinoline resistance to *E. faecalis* indicating that EfsQnr likely functioned as a DNA-gyrase inhibitor [30]. In 2009, Vetting et al. determined the three-dimensional structure of EfsQnr [17] revealing that the 211-residue protein was a PRP made up of eight complete Rfr coils composed of a mixture of type II and type IV β turns and a 12-residue C-terminal α helix capping the β helix. Two molecules of EfsQnr formed a head-to-head dimer mediated by an interaction between the C-terminal α helix of each EfsQnr molecule (Figure 5, top panel) similar to that observed in the structure of MfpA. Despite their structural similarity, a pairwise sequence alignment using EMBOSS Needle (https://www.ebi.ac.uk/Tools/psa/emboss_needle/ (accessed on 25 April 2021)) between EfsQnr and MfpA over 234 residues indicated just 19.7% sequence identity, 29.5% similarity, and 30.3% gaps. The head-to-head dimer interaction in EfsQnr involved 16 residues on one chain (chain A) and 15 residues on a second chain (chain B) involving 69 contacts encompassing an interface area of 1568 Å^2^. The head-to-head interaction was mediated similarly to in MfpA, involving interactions between parallel regions of the polypeptide backbone (residues 190–194 on chain A and 207–210 on chain B), interactions of the sidechains of the orthogonally oriented α-helices from each chain (residues 195–206 on each chain), and backbone and sidechain interactions from another short section of parallel polypeptide chains (residues 207–211 on chain A and residues 191–194 on chain B) (Figure 5, bottom panel). The hydrophobic interactions between the α-helices and adjacent strands were established by all hydrophobic, non-aromatic residues, including V94 (5 interactions), P196 (10 interactions), I200 (6 interactions), V209 (7 interactions), and I210 (7 interactions) and T211 (7 interactions) (Figure 5, bottom panel). The binding energy of the head-to-head dimer interaction was −14.9 kcal/mole, which was slightly weaker than in MfpA, that involved more residue interactions and a hydrophobic core involving aromatic side chains. 

#### 2.3.3. QnrB1 (2XTW, 2XTX, and 2XTY)

QnrB1, encoded by multi-resistance plasmids [3,31] in isolates of Enterobacteriaceae from around the world [32], is a PRP that confers moderate fluoroquinoline resistance to its host organisms. Genes encoding QnrB1 can also be found on bacterial chromosomes [32]. QnrB1 belongs to the *qnrB* subfamily of Qnr genes that include *qnrA* [3], *qnrB* [33,34], *qnrC* [35], *qnrD* [36], qnrE [37], *qnrS* [38], and *qnrVC* [39,40] subfamilies [41]. 

The three-dimensional structure of QnrB1 from *K. pneumoniae* was determined by Vetting et al. in 2011 [18]. The 226-residue protein contained nine Rfr coils composed of a mixture of type II and type IV β turns, a 12-residue α helix capping its C terminus (D197–L208) and two loops projecting outward from the Rfr coil (an 8-residue loop (Loop A: Y46-G53) in coil 2 connecting face 2 to face 3 and projecting outward from face 2, and a 12-residue loop (Loop B: S102–S113) that projects outward from the corner between face 4 and face 1 joining coil 4 and col 5) (Figure 6, top panel). The head-to-head dimer was established by the interaction of the C-terminal helix of two QnrB1 molecules mediated by 40 non-bonded contacts involving 23 residues (chain A: 13 residues and chain D: 10 residues) for the interaction between chain A and chain D and 28 residues between chain B and chain C (chain B: 14 and chain C: 14) for the two dimers observed in the crystallographic asymmetric unit. The hydrophobic core was established by interactions between non-aromatic sidechains from each α helix, including I186 (five interactions), M205 (eight interactions), and I210 (eight interactions) (Figure 6, bottom panel). The interface between the A and D chains had an area of 1131 Å^2^, whereas the interface between the B and C chains had an area of 1432 Å^2^. Despite the rather large difference between the interface areas, the interaction energies between two dimers composed of different pairs of chains in the asymmetric unit were quite similar with an average of −10.6 kcal/mole. The dimer interface in QnrB1 involved fewer residues (23) and fewer contacts (40) and weaker binding (−10.6 kcal/mole) compared to MfpA (35 residues, 81 contacts and −15.8 kcal/mole) and EfsQnr (31 residues, 69 contacts and −14.9 kcal/mole). The authors noted potentially interestingly positioned residues in the two loops that were also conserved, indicating that they may define a possible contact surface with topoisomerases to assist binding with gyrase in addition to the guiding electrostatic interactions [18]. The authors also pointed out that deletion of the smaller loop affected gyrase inhibition while loss of the larger loop resulted in total inability to inhibit gyrase [18]. 

In 2019, Li et al. showed that QnrB increased bacterial mutation rates and the selection of quinolone-resistant mutants [33]. Transcriptomic and whole genome sequencing analysis indicated that QnrB upregulated gene expression and increased the number of gene copies near the origin of replication in both *E. coli* and *K. pneumoniae* [33]. The authors also reported that Bacterial two-hybrid and in vitro pull-down assays indicated that QnrB interacted with the DNA replication initiator DnaA [33]. 

#### 2.3.4. AhQnr (3PSS and 3PSZ) 

Xiong et al. reported the structure of AhQnr, a Qnr-like protein from *A. hydrophila* in, 2011 [20]. AhQnr adopted a Rfr fold with nine complete coils composed of a mixture of type II and type IV β turns capped by a 10-residue α-helix (D101-I210) and two conserved loops (Loop A (F47-C56) and Loop B (F103-C114) (Figure 7, top panel). The overall structure of AhQnr was very similar to that of QnrB1, consistent with the 38.8% sequence identity plus 56% similarity. The dimer interaction between two AhQnr chains involved a common motif with a short parallel β-sheet formed by residues Q197-N199 of chain A with I213-F215 of chain B, sidechain interactions between the two orthogonally-oriented α-helixes from each molecule, and a final short parallel β-sheet formed by L212-F215 of chain A with Q197-N199 of chain B with the interaction mediated by 96 contacts involving 20 residues on chain A and 20 residues on chain B. Interestingly, the dimer interaction was established by an equal mixture of polar residues and eight hydrophobic residues, with Q23 and F215 making the largest numbers of contacts at 11 and 12 interactions, respectively (Figure 7 bottom panel). The number of residues establishing the dimer interface in AhQnr (40) was greater than in MfpA (35), EfsQnr (31), and QnrB1 (23) and, correspondingly, the interface area of AhQNr (2039 Å^2^) was significantly larger than that of MfpA (1700 Å^2^), EfsQnr (1568 Å^2^) and QnrB1 (1131–1432 Å^2^). Interestingly, despite having the largest interface area, the energy of the AhQNr dimer interaction was the weakest of those discussed so far at −9.9 kcal/mole, compared to MfpA (−15.8 kcal/mole), EfsQnr (−14.9 kcal/mole), and QnrB1 (−10.6 kcal/mole). It is possible that more interactions were required to compensate for a less stable hydrophobic core, with the aromatic sidechain involved in the largest number of contacts, F215, residing on the surface of the protein and not participating in formation of a hydrophobic core. 

#### 2.3.5. AlbG (2XT2 and 2XT4)

AlbG is a self-resistance factor from *X. albilineans* against albicidin, a nonribosomally-encoded hybrid polyketide-peptide with antibiotic and phytotoxic properties produced by the pathogenic bacterium *X. albilineans* [42]. As a self-resistance factor, AlbG protects *X. albilineans* from the antibiotic and cytotoxic effects of albicidin produced by *X. albilineans* itself. Albicidin shares a common mode of action with fluoroquinolines, also stabilizing the DNA gyrase-cleaved DNA complex and leading to single and double-strand breaks and eventual cell death [42]. *X. albilineans* uses multiple self-protection mechanisms, including the DNA mimicry of the AlbG PRP. AlbG increases resistance of the *E. coli* gyrase to albicidin both in vivo and in vitro, and at higher concentrations it inhibits supercoiling by the *E. coli* gyrase even in the absence of albicidin [42]. 

The three-dimensional structure of AlbG was solved by Vetting et al. in 2011 [19]. The structure is similar to other Qnr proteins, having eight complete coils composed of a mixture of type II and type IV with one half coil as the zeroth coil and a quarter coil as ninth coil, capped at the N-terminus of the Rfr helix by a small N-terminal extension (residues 1–8) and capped at the C terminal end of the Rfr helix by an 11-residue α helix that is involved in the formation of the head-to-head functional dimer (Figure 8, top panel). A 13-residue loop insertion (T87-A99) disrupted the β helix structure of AlbG, with the loop and β-helical kink stabilized by several noncanonical PRP residues [19]. The dimer interface established by the head-to-head interaction of the C-terminal helix of two AlbG molecules involves 36 amino acids (chain A: 18 residues and chain B: 18 residues) that involved the formation of a hydrophobic core involving eight aromatic and aliphatic sidechains from each chain and interactions between five charged sidechains from each chain (Figure 8, bottom panel). The surface area for the dimer interface was 911 Å^2^ and the binding energy for dimer formation was −13.5 kcal/mol. 

#### 2.3.6. PENT (6FLS)

In 2018, Notari et al. reported the three-dimensional structure of PENT [43], a PRP from the human pathogen *Clostridium botulinum,* which is a homolog of the fluoroquinoline resistance protein, MfpA from *M. Typhimurium* (20.2% sequence identity and 39.3% sequence similarity over 183 residues). PENT is made up of 217 residues, crystallizes as a dimer and adopts a right-handed quadrilateral β helix with eight coils composed of a mixture of type II and type IV β turns, with a 12-residue α helix with its axis perpendicular to the β helix axis capping both the N- and C-termini of the β helix (Figure 9). The head-to-head dimer interface is mediated by the C-terminal α helix cap of each of two PENT molecules (Figure 9, top panel). The head-to-head dimer interface interaction was observed between chains A and B and between chains C and D in the crystallographic asymmetric unit. The architecture of the dimer interface involved the formation of a short parallel β involving A197-I199 on chain A and I213-V215 on chain B, orthogonally oriented α helices (S200-G212 on each chain), and another short parallel β sheet formed by residues I213-V215 on chain A and residues A197-I199 on chain B (Figure 9, bottom). The dimer interaction involved 37 residues (chain A: 18, chain B: 19) and 63 non-bonded contacts mediated by a network of aromatic and aliphatic sidechains that formed a hydrophobic core with I199, M201, I213, and I214 all making six or more contacts in chain A and with S189, I199, W211, I213, and I214 making at least six contacts in chain B (Figure 9, bottom panel). The interface area of those two interactions was the same at ~1035 Å^2^ and the energy of the interaction was −20 kcal/mol. Although the PENT PRP forms a head-to-head dimer with an overall structure similar to that of known PRP gyrase inhibitors, i.e., the MfpA and EfsQnr, it lacked the conserved extra-helix loops observed in other PRP gyrase inhibitors, such as QnrB1, AhQnr, and AlbG. At this time, the function of PENT has not been confirmed experimentally.

### 2.4. Ubiquitin E3 Ligases

#### 2.4.1. SopA (2QYU, 2QZA, 3SY2 and 5JW7)

*Salmonella* enterica serovar *typhimurium*, a rod-shaped, flagellate, facultative anaerobic, Gram-negative pathogenic bacterium, stimulates inflammatory responses in the intestinal tract that are required in order for it to replicate in the intestinal tract [44]. Bacterial pathogens can infect host cells and stimulate the inflammatory response by delivering effector proteins by either a type III or type IV secretion system [45]. *S. Typhimurium* uses a type III secretion system to deliver effector proteins to its host cells [46]. One of the effector proteins is SopA, which is a Homologous to E6AP C-terminus (HECT)-type E3 ligase [47] that is required for efficient stimulation of inflammation in *S. Typhimurium* infections [48]. The process of attaching ubiquitin to a targeted protein requires an enzyme cascade including ubiquitin activation enzyme (E1), conjugating enzymes (E2), and ligase (E3). HECT E3s are of two types, with one E3 functioning by forming a thioester intermediate with ubiquitin to transfer ubiquitin to substrate and the other is called a really interesting new gene (RING) E3. Bacterial infection is normally sensed by host pattern recognition receptor-mediated detection of pathogen-associated molecular patterns (PAMPS), which induces a pro-inflammatory response to fight the infection [49]. The tripartite motif-containing (TRIM) TRIM56 and TRIM65 host RING [50] E3 ubiquitin ligases are normally involved recognizing foreign proteins and stimulating release of interferons to communicate to nearby cells to launch an immune response to combat infection [49]. Kamanova et al. showed that SopA inhibits the host immune responses by targeting the TRIM56 and TRIM65 host RING [50] E3 ubiquitin ligases [51]. 

In 2008, Diao et al. solved the structure of SopA_163–782_, a fragment of the full-length SopA that was stable to proteolysis [44]. The structure was described as being organized into a 147-residue N-terminal β-helix domain (residues 163–370), a central domain (residues 371–590), a helical linker (residues 591–611) and a C-terminal domain (residues 612–782). SopA was the first PRP-containing multi-domain protein to have its structure determined. The SopA structure contains an N terminal PRP domain (around 200 amino acids) and a catalytic domain containing N- and C-lobes (Figure 10, top panel). The Rfr domain in SopA is comprised of four complete coils composed of a mixture of type II and IV β turns. In face 2 of the first coil, the β helix is replaced by two α-helixes. One is composed of five residues from T206 to I210 and the other is made up of six residues including P212 to E217. After the last complete Rfr coil, there is an extended quarter coil connected with the linker of β-helix domain and central domain. In 2017, Fiskin et al., solved another structure of SopA bound to the RING domain of TRIM56 [49] which revealed the structural basis for selectivity of SopA for TRIM56 and TRIM6 showing that the TRIM56 domain interacts with the interface of the β-helix and the N-lobe domains of SopA through packing with the first Zn^2+^-binding loop in a cleft of SopA. Using a combination mutation experiments, it was shown that this interaction relied on three key residues including T338 of SopA, and L25 and E26 of TRIM56. In TRIM56, E25 interacts with R296, H297, and K298 via polar contacts while L25 is inserted between the hydrophobic pocket constructed by P334 and F345. In SopA, T338 has a close hydrophobic contact with central α-helix of TRIM56. From the mutation experiments, T338 was also shown to support the interaction between SopA and TRIM65 (Figure 10, bottom panel) [49]. The authors performed structure-based biochemical analyses to show that SopA inhibited the TRIM56 E3 ligase activity by occluding the E2-interacting surface of TRIM56, and further showed that SopA ubiquinates TRIM56 and TRIM65 resulting in their proteasomal degradation during infection, thus disrupting the host immune response to *S. typhimurium* infection. 

#### 2.4.2. NleL (3NB2, 3NAW, and 3SQV)

The non-Lee-encoded effector ligase (NleL) from enterohemorrhagic *Escherichia coli* (EHEC) 0157:H7 is a homolog of SopA from *S. Typhimurium*. Lin et al. [52] solved the crystal structure of NleL and showed that NleL functionally and structural mimics eukaryotic E3 ligases and catalyzes formation of unanchored polyubiquitin chains using linkages with residues Lys6 and Lys 48 and with the catalytic cysteine residue forming a thioester intermediate with ubiquitin [52]. In recent studies, it has been shown that NleL uses JNK proteins (stress-activated protein kinase) as the first substrate to promote EHEC-induced attaching and effacing (A/E) lesions [53] and interacts with TRAF2, TRAF5, TRAF6, IKKα, and IKKβ to disrupt the host NF-κB pathway [54]. 

The structure of NleL shares a common N-terminal PRP domain with SopA [52,55], which contains five and a half Rfr coils mostly composed of type II β turns with one type IV β turn from L213 to L216, one α helix at the N terminus and three α helices at the C terminus flanking the β helix domain. Superposition of the SopA and NleL crystal structures indicated that the PR and N-lobe domains maintain a fixed relative orientation, the C-lobes can adopt different orientations relative to the PR and N-lobe, perhaps but due to their different functional requirements [52]. Based on the structures of complex between NleL and the UbcH7 domain (3SQV) [55], UbcH7 contacts the N-lobes by both hydrogen bond and van der Waals interactions. In SopA, the PRP domain is N-terminal to the catalytic domain and the carbohydrate-modified substrate proteins may be recognized by that. However, the actual functions and substrates still remain unknown. Although the function of PR domains is still unknown, a cleft at the interface of PRP domain and N-lobe domain provide some clues to identify the potential functions. Two of tripartite-motif-containing (TRIM) E3 ligases, belonging to the family of RING-type E3 ligases, are involved in the regulation of SopA [51]. Complex structure involving SopA and TRIM 56 unmasked that the first Zn^2+^-binding loop of TRIM 56 contacts with the cleft of SopA. In TRIM 56, Leu25 and Glu 26 are the two key amino acids contacting SopA even though they adopt different strategies. Leu25 interacts with Phe345 and Pro334 of SopA by inserting into a hydrophobic pocket while Glu26 contacts Arg296, His297, and Lys298 by their polar groups. (Figure 11). There are two complexes in the crystallographic asymmetric unit. In the interaction between chain A (NleL) and chain C (E2 UbcH7), one salt bridge, two hydrogen bond and 47 non-bonded contacts form the contact interface involving 15 residues from chain A and 13 residues from chain C. Asn578 from NleL and Phe63 from E2 UbcH7 are responsible for most of the contacts for this interaction. The interface area between two chains was 1640 Å^2^ with the energy of interaction at −11.5 kcal/mol. Different from interaction between chain A and C, in the interaction between chain B and chain D, a disulfide bond between Cys753 (chain B) and Cys86 (chain D) stabilized the interaction. Twenty residues from chain B and 15 residues from chain D formed the interaction and the interface area was 1922 Å^2^ with an energy of formation of −15.3 kcal/mol. Phe63 from chain D was major contributor to support this interaction.

### 2.5. Synaptic Vesicle Glycoprotein 2 Receptors

#### SV2C-LD (4JRA, 5JMC, 5JLV, 5MOY, and 6ES1)

Synaptic vesicles, also referred to as neurotransmitter vesicles, store various neurotransmitters that are released at the synapse, i.e., the junction, or between nerve cells. The synaptic vesicles are essential for propagation of nerve impulses and constantly regenerated in nerve cells. The synaptic lumen refers to the volume contained inside the synaptic vesicles. Synaptic vesicle glycoprotein 2 (SV2) receptors represent a protein family with two essential complementary major isoforms, SV2A and SV2B, and one minor isoform, SV2C, that are putative transport proteins [56]. All three isoforms are composed of a 12-transmembrane domain and a luminal domain (SVC-LD), i.e., the domain sticks into the vesicle lumen, composed of a four and half-coil PRP β helix that acts as a receptor for binding to Botulinum neurotoxin A (BoNT/A) from the bacterium *C. botulinum* and related bacteria [57]. Botulinum neurotoxins (BoNTs) are the most toxic class of bioweapons and also have a popular and widely used cosmetic application as an anti-wrinkle agent, e.g., Botox [58]. BoNTs exist as seven main serotypes from BoNT/A to BoNT/G [58]. In 2006, Dong et al. showed that the luminal domain of SV2 (SVC-LD) acts as a receptor for BoNT [57]. The SV2C-LD is necessary in the process of translocation of BoNT/A and glycosylation of SV2C-LD and SV2 glycan are also crucial for BoNT/A binding to neurons [58].

From 2014 to 2018, five structures were reported for complexes between BoNTs and the SV2C-LD (also referred to as SV2C-L4) [58,59,60,61]. The LD of SVC2 is a five-coil PRP domain composed exclusively of type IV β turns (Figure 12). All five structures were similar with slight variations in the relative orientations between the SV2C-LD and the different subtypes of BoNT/A. The interaction between the BoNT/A and the SV2C-LD receptor occurs at the exposed β-strand of the 5th Rfr coil at C-terminal un-capped edge of the PRP domain by forming an interchain β-sheet mediated by backbone-to-backbone hydrogen bonds between the open β strand of SV2C-LD and the β-strand edge of the BoNT/A. Even though the structural features of the multiple SV2C-LD/BoNT/A complexes are similar, the orientation of the BoNT/A relative to the SV2C-LD β-helix receptor varied slightly among the structures due to differences in amino acids sidechains mediating the binding interaction and due to flexibility of the interactions [61]. For example, in two structures reported for the same complex SV2C-LD/BoNT/A but crystallized in different space groups and with different resolution (PDB-IDs 5MOY and 6ES1), the orientation of the SV2C-LD β-helix was rotated by 15° relative to BoNT/A to maintain the tight interaction between β-hairpin of BoNT/A and the continuing β-sheet [61]. This rotation caused shifts in both components. The residues in the β-hairpin (T1146 and N1147) moved by 1.8 Å and the residues from C-terminus of SV2C-LD (N480 to Y497, D546 to K566) moved between 0.4 and 8.1 Å [61]. 

## 3. Cyanobacterial and Plant PRPs with Three-Dimensional Structures but Unknown Function

### 3.1. HetL (3DU1)

HetL (gene all3740) is one of more than 30 PRPs from *Nostoc* sp. PCC 7120 [12]. In 2002, it was shown that HetL overexpression using a heterologous promoter in wild-type *Nostoc* PCC 7120 induced multiple-contiguous heterocysts in nitrate-containing medium [62]. Addition of a synthetic peptide containing the last five residues of PatS, which was known to suppress heterocyst differentiation in wild type *Nostoc* PCC 7120, did not suppress heterocyst differentiation in the *hetL* overexpression strain, indicating that HetL acts downstream of PatS production. Interestingly, a *hetL* null-mutant showed normal heterocyst development and diazotrophic growth, i.e., the ability to fix atmospheric nitrogen into more usable forms such as ammonia, leading Liu and Golden to conclude that HetL may not normally involved in regulating heterocyst development, many only play a non-essential accessory role, or that its function may be compensated for by cross talk or redundancy with other PRPs [62]. Liu and Golden observed that the predicted HetL protein was composed almost entirely of PRs. 

In 2009, the three-dimensional structure of HetL was determined, representing the first PRP structure from *Nostoc* sp. PCC 7120 [12]. The structure revealed that HetL adopted the standard right-handed quadrilateral β helical structure composed of ten complete coils entirely composed of type II β turns, a ten-residue α helix that caps its N terminus, a two-stranded anti-parallel β sheet that sits on the C-terminus of Face 1 of the β helix, a six-residue loop insertion protruding from the corner adjoining Face 3 and Face 4 near the middle of the helix, and a nine-residue insertion loop protruding from the corner joining Face 3 and Face 4 in the C-terminal half of the helix (Figure 13). The electrostatic surface potential of HetL contained patches of negative charge but is otherwise unremarkable. Although HetL has been shown to play a role in heterocyst differentiation, its precise biochemical function remains unknown.

### 3.2. Alr1298 (6UV7 and 6UVI)

The full-length Alr1298 protein is predicted to contain 167 amino acids, which includes 15 PRs. In 2020, Zhang et al reported the three-dimensional structure of Alr1298 [63], which was found to have three and three quarter Rfr coils composed of a combination of type II and type IV β turns with the incomplete coil occurring at the C-terminus of the β helix. (Figure 14). The β helix was capped at the C-terminus by a single α helix and a five α helix bundle at its N-terminus. The electrostatic surface potential contained large patches of clustered positive and negative charge that are poised to interact with charged binding partners. Potential clues regarding the function of Alr1298 were investigated by analyzing the gene cluster to which the alr1298 gene belonged since the genes in a cluster possibly belonged to a common operon that often share related functions [64,65,66]. The gene cluster contains three genes preceding and three genes following *alr1298*. Alr1295 was found to be conserved in 14 of 15 aligned genomes and encoded a prohibitin, which generally act as inhibitors to cell proliferation. In cyanobacteria, prohibitins have been linked to thylakoid biogenesis and membrane synthesis. Alr1297 was annotated as an ABC transport system. Alr1299 was predicted to be involved in purine metabolism, metabolic pathways and biosynthesis of secondary metabolites. The other genes were unannotated. Given the proximity of the *alr1298* gene to an annotated prohibitin, it is possible that Alr1298 plays a role in cell proliferation and thylakoid biogenesis. A genome-wide microarray analysis revealed that alr1298 was upregulated following nitrogen starvation, ref [67] peaking at a 4x increase at eight hours post nitrogen starvation. Since the primary response to nitrogen starvation is patterned differentiation of vegetative cells into heterocysts capable of fixing atmospheric nitrogen, the microarray result supports the observation that alr1298 may either be involved in the response to nitrogen starvation or play a role in heterocyst differentiation [63].

### 3.3. Alr5209 (6OMX)

The *alr5209* gene from *Nostoc* sp. st. PCC7120 encodes a 129 amino acid protein that contains 16 tandem PRs [21]. Zhang et al. determined the three-dimensional structure of Alr5209 in 2019 [21] revealing that it was composed of four complete Rfr coils of a β helix that was capped by a nine-residue α helix at the N-terminus and by a four-reside α helix at the C-terminus (Figure 15). Alr5209 was the first PRP identified to contain type I β turns in its β helix structure, with all four Rfr coils joining Face 2 to Face 3 being type I β turns and all the remaining 12 turns being type II β turns. In comparison with other PRPs, involvement of type I β turns in Alr5209 resulted in a more compact structure compared to PRP structures composed of a combination of type I and type II β turns used to form β coil stack. Analysis of the electrostatic surface potential revealed that two faces of the β helix were predominantly negatively charged with the other two faces being of mixed charge and generally neutral overall. Analysis of the gene cluster that alr5209 belonged to indicated that it may play a role in oxidative phosphorylation [21].

### 3.4. Np275/Np276 (2J8K and 2J8I)

The *Np275* and *Np276* genes in *Nostoc punctiforme,* which are adjacent to one another and encode proteins of 98 and 75 amino acids, respectively, have sequences that are composed of tandem PRs [13]. The structure of Np275 was solved at 2.1 Å resolution (2J8I). The majority of the Np275 structure adopts a Rfr fold composed of four complete coils will all type II β turns and with the N-terminal end being capped by an α helix and the C-terminal end of the coil being uncapped exposing the hydrophobic core and terminal β strands of the β helix. The intervening sequence between the stop codon of the *Np275* gene and the start codon of *Np276* gene also encoded an in-frame PR sequence suggesting that *Np275* and *Np276* previously existed as a single longer protein. This suggestion was supported by the fact that it was possible to solve the structure of a Np275-Np276 fusion protein composed of seven and three-quarters complete Rfr coils with the N-terminal Np275 portion having virtually the same structure as the Np275 monomer structure and the Np276 portion also being uncapped and with the entire Rfr coil composed of type II β turns (Figure 16). Interestingly, the authors noted that Np275/Np276 has an unoccupied internal molecular surface along the Rfr coil helical axis that is continuous with a volume of 281 ${Å}^{3}$, and pointed out that while the function of MfpA is related to glycolipid localization, the cavity in Np275/Np276 would not be large enough to accommodate the hydrophobic tail of a glycolipid without expansion [13]. A putative function of the tunnel in Np275/Np276 remains unknown. 

### 3.5. Rfr32 (2F3L and 2G0Y)

The *Rfr32* gene from *Cyanothece* sp. 51142 encodes a 167-residue that includes a 29-residue N-terminal signal peptide [14]. Buchko et al. determined the three-dimensional structure of Rfr32 minus the N-terminal 29 residue signal peptide at 2.1 Å in 2006 revealing a structure dominated by five and one-quarter uninterrupted Rfr coils (Figure 17). The C-terminus of the Rfr coil is capped by a two-α helix bundle that is stabilized by an internal disulfide bond. The Rfr coil of Rfr32 contains mixture of type II and IV β turns. The electrostatic surface potential of Rfr32 contains contiguous patches of negative charge on Face 3 and in a deep crevasse present on Face 4. Rfr32 is predicted to reside in the thylakoid lumen (https://www.uniprot.org/uniprot/B1WVN5#subcellular_location (accessed on 25 April 2021)). The function of Rfr32 remains unknown and the UniProt database reports that existence of the protein is predicted based on homology (https://www.uniprot.org/uniprot/B1WVN5 (accessed on 25 April 2021)).

### 3.6. Rfr23 (2O6W) 

In 2008, Buchko et al. determined the three-dimensional structure of Rfr23, the second PRP with known structure from *Cyanothece* sp. 51142 [15]. The Rfr β helix contained five complete coils composed exclusively of type II β turns with one helix in the N terminal (Figure 18). Different from Rfr32, there are two distinctive structural features in Rfr23, one is a 24-residue insertion between the conjunction of the first and secondary coil which causes a break of consensus sequence. Due to the missing electron density, the structure of this insertion still remains unknown. However, according to the analysis of sequence, this insertion has a positive charge. Moreover, the formation of disulfide bracket between Cys39 and Cys42 make this structure more stable and it is possible to contribute the activity of Rfr23. While the function of Rfr23 remains unknown, the UniProt database indicates that experimental evidence exists for expression of Rfr23 at the protein level (https://www.uniprot.org/uniprot/D0VWX3 (accessed on 25 April 2021)).

### 3.7. At2g49920.2 (3N90)

The genome of *A. thaliana* is predicted to contain three genes encoding PRPs (At2g44920, At5g53490 and At1g12250) all of which are predicted to be located in the thylakoid lumen [68,69]. In 2011, Ni et al. reported the three-dimensional structure of At2g49920.2, one of two isoforms of At2g49920 identified in *A. thaliana* [16]. At2g49920.2 contained five complete Rfr coils composed exclusively of type II β turns with one gamma turn (Figure 19) made up of contained 25 uninterrupted PRs with a single-turn α helix capping the N-terminus and two α helices stabilized by a disulfide bond capping the C-terminus of the β helix. Although the function of At2g49920.2 is still unknown, the chloroplast thylakoid lumen, in which At2g49920.2 is predicted to be located, is a compartment where the reactions of oxygenic photosynthesis take place. It has also been shown that At2g49920.2 is primarily expressed in the leaves of *A. thaliana* [70].

### 3.8. Changes in PRP Gene Expression Levels Nostoc sp. st. PCC 7120 in Response to Nitrogen Deprivation

In 2006, Ehira and Ohmori published a genome-wide gene expression experiment using a *Nostoc* (Anabaena) microarray containing 5336 probes specific for genes on the chromosome whose expression levels changed in response to nitrogen deprivation at 3, 8, and 24 h post nitrogen deprivation compared to at the 0 h timepoint [67]. Of the 26 chromosomally encoded PRP genes, expression of 25 of the genes was detected in the microarray analysis, 15 experienced at least a twofold increase in expression following nitrogen deprivation for at least one of the time points (Figure 20). By comparison, expression of only two of the 25 genes decreased by at least 40% (Figure 20).

## 4. Discussion

Despite their intriguing structure, and variations therein, the large size of their Pfam superfamily with nearly 39,000 members, and their relative abundance in one of the most ancient and important organisms on earth, i.e., oxygenic cyanobacteria, the biochemical function of PRPs remains remarkably elusive. To date, there are only two examples where the explicit biochemical function of a PRP is known, as discussed in this review. The first cellular function being that of conferring antibiotic resistance to fluoroquinoline antibiotics through the biochemical function of acting as DNA gyrase inhibitors, such as MfpA and the Qnr family of proteins, that exert their function by acting as a DNA mimic, binding to the complex of DNA gyrase and DNA, and blocking binding of fluoroquinoline to the DNA gyrase DNA complex, and therefore blocking its antibiotic activity. The second clear function is seen in SV2C that functions as a BoNT/A receptor in the synaptic vesicles of neurons. In this activity, the BoNT/A toxin binds to the SV2C PRP luminal domain of a synaptic vesicle neurotransmitter membrane protein, with the binding interaction mediated by a dovetailing of the β-strands of the BoNT/A neurotoxin with an exposed β-strand edge of the PRP luminal domain, resulting in the formation of an extended β-sheet that spans and crosses over the PRP luminal domain and the BoNT/A neurotoxin molecule. This completes the list of examples for which a PRP or a PRP domain of a larger protein is known to carry out a specific biochemical function that directly involves the PRP structure itself. From there, our understanding of PRP function becomes less clear. We have seen that PRP domains are present in the SopA ubiquitination inhibitor whose biochemical function is to bind to TRIM56 and TRIM65 and block the host immune response of interferon production that would normally stimulate proteasome targeting of the bacterial proteins as part of the host immune response to infection; however, the PRP domain of SopA does not directly interact with the targeted TRIM proteins leaving the precise function of the PRP domain of SopA in question. From here, a biochemical function has been associated with a few other PRPs, e.g., the prototypical HglK protein was associated with localization of glycolipids to the heterocyst outer layer, but the structure of HglK is unknown and the precise biochemical function of HglK remains unknown. The same is true for RfrA, for which a putative role in regulating an uncharacterized manganese uptake system was proposed, but the structure of RfrA and the nature of the putative manganese uptake system remains uncharacterized. Similarly, HetL has been shown to play a role in regulating heterocyst differentiation, and the structure of HetL has been determined, but no connection between the structure and the proposed biochemical function has been elucidated. The remaining structure/function space of the PRP superfamily remains completely uncharacterized.

What we can deduce at this point regarding the structure and function of PRPs, and multi-domain proteins containing PRP domains, is that in some cases the function is a consequence of the shape and electrostatic surface potential of the PRP, as is observed in the case of DNA mimicry in MfpA [6] and the Qnr family of proteins [3,30,34,35,37,38,39,40,41]. In other cases, the PRP domain makes a direct interaction with another protein to mediate its function, such as with the SV2C-LD binding to BoNT/A [58,59,60,61]. In other cases, the PRP domain simply acts as a scaffold to provide a surface to support binding interactions with another protein, as was observed in the ubiquitin E3 ligases [44,49]. We have seen that variations on the PRP scaffold structure can play functional roles, such as the extra-β-helix loop excursions observed in some Qnr-family proteins, such as QnrB1 [18,33] and AhQnr [20], which appear to play critical roles in guiding interactions with the DNA gyrase. We have also observed many subtle variations in Rfr fold structures that may turn out to be important to function, including small bulges that project from the β-helix structure, variations in the compositions of the β-turns, i.e., the mixture of type I, type II, and type IV β turns, which cause subtle changes in the β-helix dimensions or β-helix twist [21], as well as the presence or absence of N-terminal or C-terminal capping α-helices, and whether or not the PRP constitutes a domain in a multidomain protein. 

In closing, while the structure-space of PRPs becomes richer, our understanding of the biochemical function of members of the PRP superfamily lags increasingly behind. Targeted and carefully designed studies are required to begin to chip away at expanding our understanding the repertoire of structures and functions of the enigmatic members of the PRP superfamily.

## Figures and Tables

**Figure 1 biomolecules-11-00638-f001:**
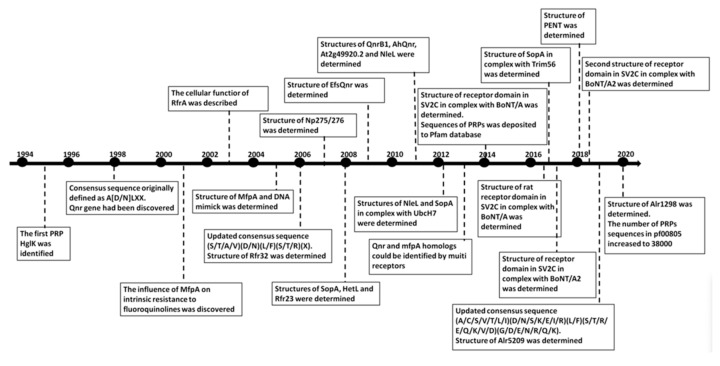
Milestones in the timeline investigation of the structure and function of PRPs.

**Figure 2 biomolecules-11-00638-f002:**
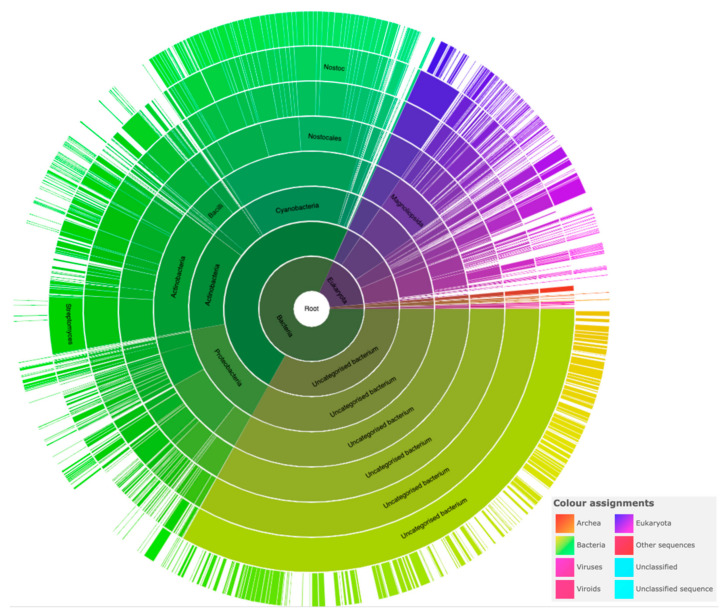
Distribution of PRP sequences across species. This sunburst plot of the PF00805 PRP Pfam shows the distribution of 38,981 sequences across 3338 species. The color-coding in the sunburst plot is indicated in the legend.

**Figure 3 biomolecules-11-00638-f003:**
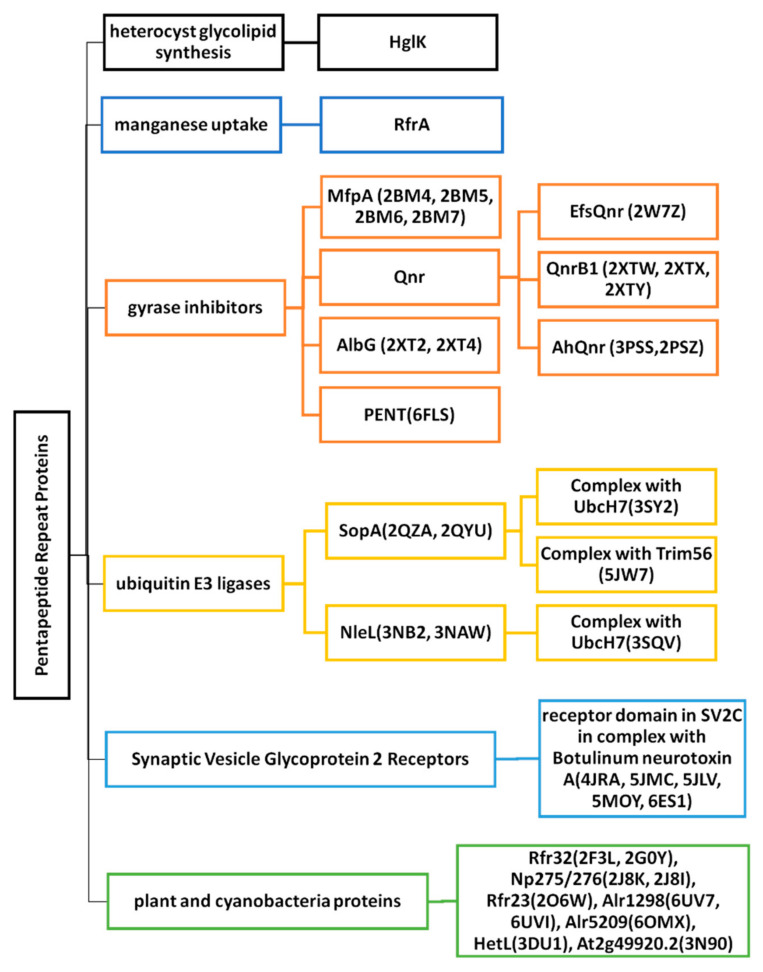
Summary of the PRPs discussed with and without known three-dimensional structures. The six category groups are shown in the first branch. The second and subsequent branches indicate specific PRPs. PRPs with known structures include the corresponding PDB ID inside parentheses immediately to the right of the PRP name.

**Figure 4 biomolecules-11-00638-f004:**
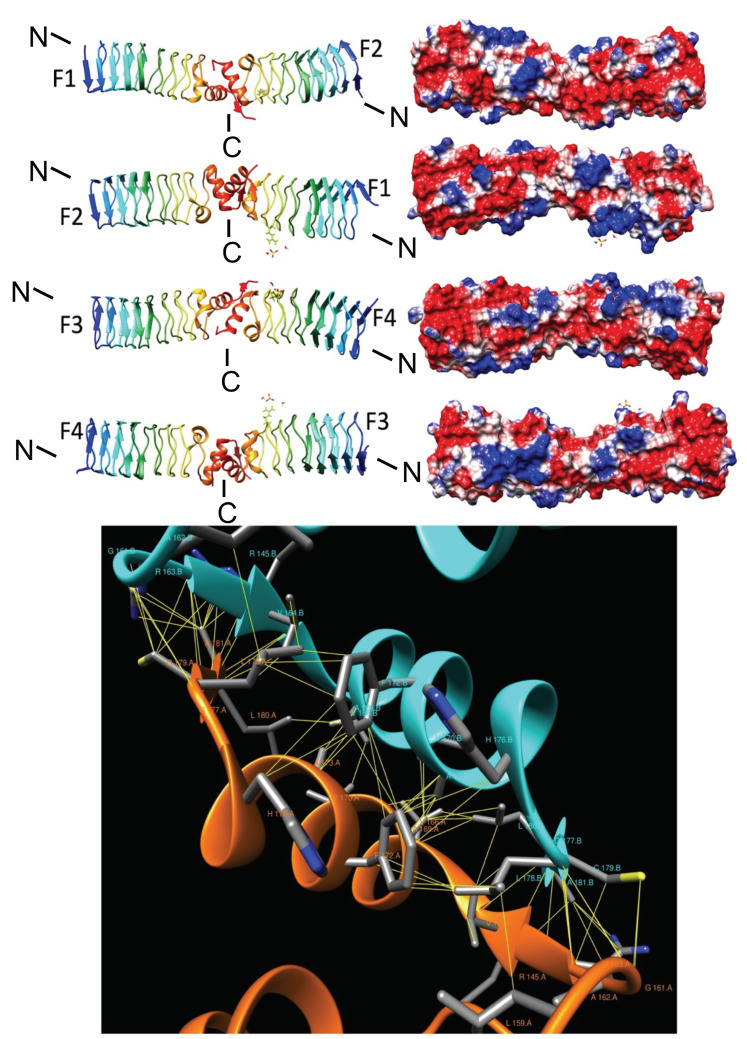
Structure and analysis of MfpA. The top panel (at left) shows the ribbon diagrams looking at the four different faces of the Rfr coil the structure and the corresponding electronic potential surface analysis (at right). The bottom panel shows the head-to-head dimer interface with non-bonded interactions depicted by yellow lines as identified using Intersurf [28] and depicted using Chimera [29]. Chain A is colored orange, Chain B is colored cyan, and interacting residues are labeled.

**Figure 5 biomolecules-11-00638-f005:**
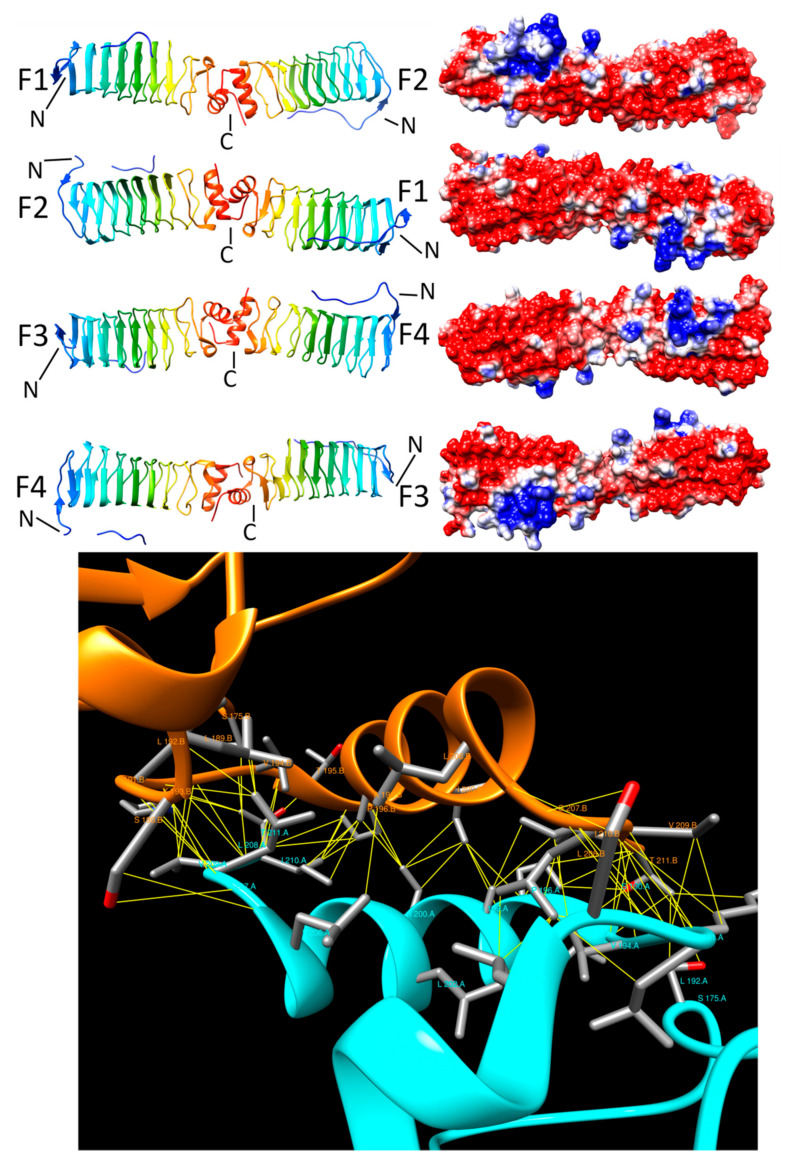
Structure and analysis of EfsQnr. The top panel shows the ribbon diagrams of the structure for four different orientations (at left) and the corresponding electronic potential surface plots (at right). The bottom panel is the interface model calculated by Intersurf and depicted using Chimera.

**Figure 6 biomolecules-11-00638-f006:**
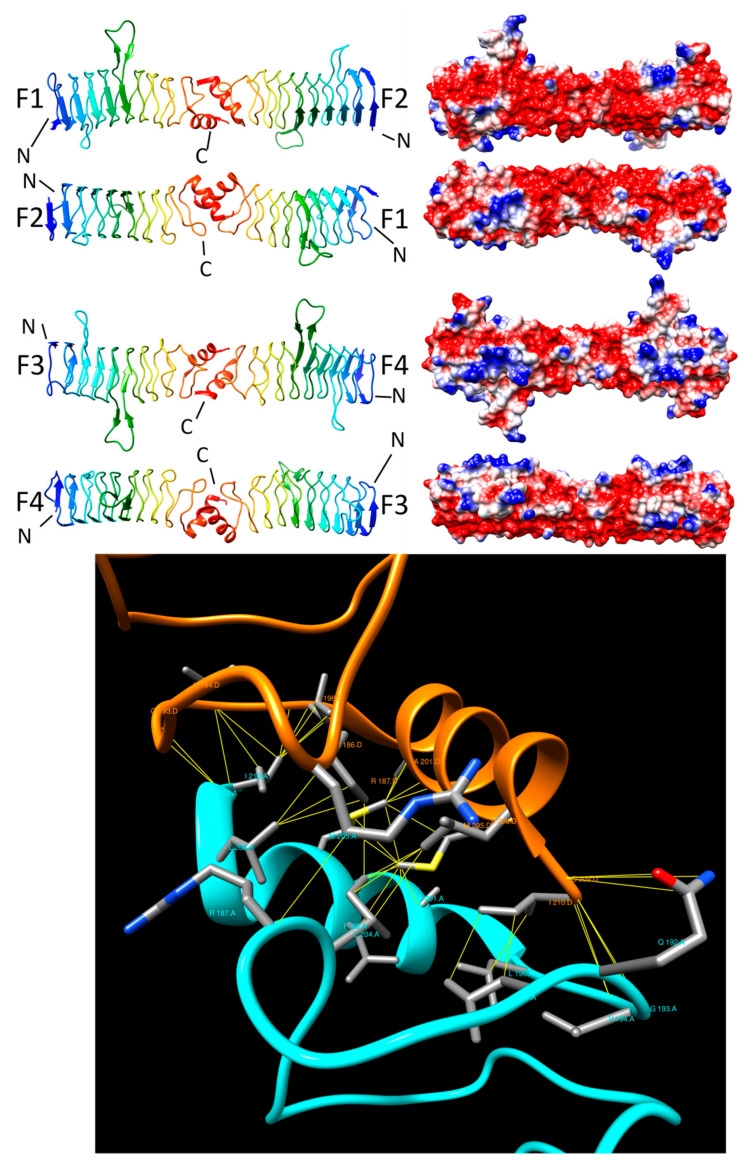
Structure and analysis of QnrB1. The top panel shows the ribbon diagrams of the structure for four different orientations (at left) and the corresponding electronic potential surface plots (at right). The bottom panel is the interface model calculated by Intersurf and depicted using Chimera.

**Figure 7 biomolecules-11-00638-f007:**
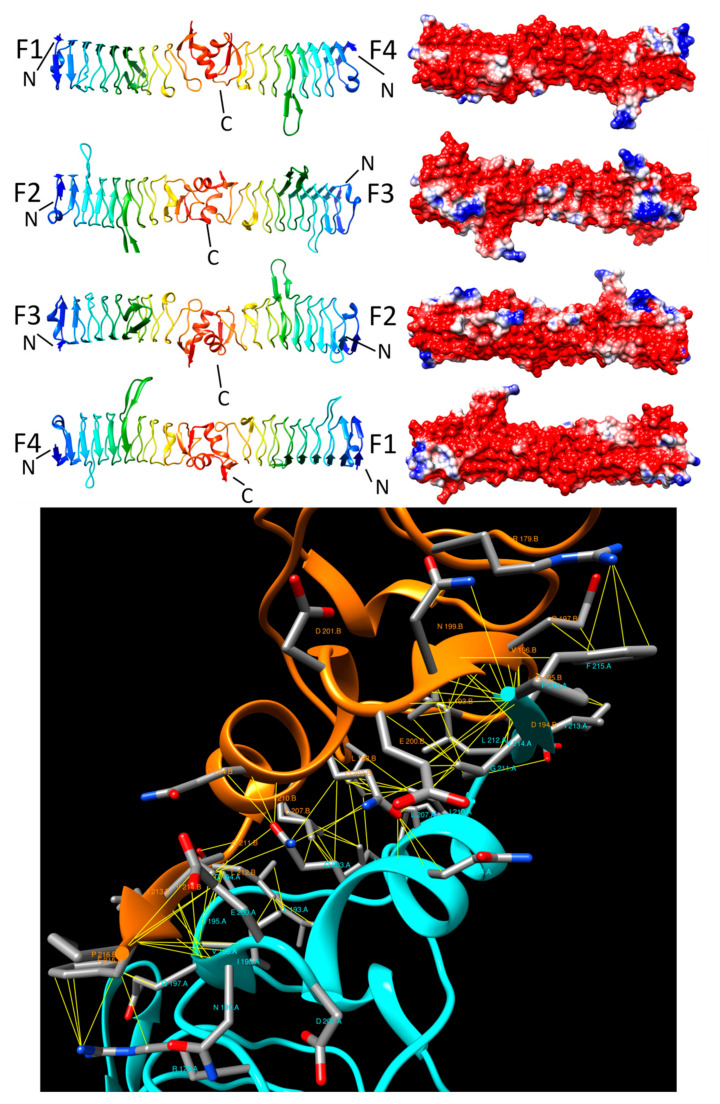
Structure and analysis of AhQnr. The top panel shows the ribbon diagrams of the structure for four different orientations (at left) and the corresponding electronic potential surface plots (at right). The bottom panel is the interface model calculated by Intersurf and depicted using Chimera.

**Figure 8 biomolecules-11-00638-f008:**
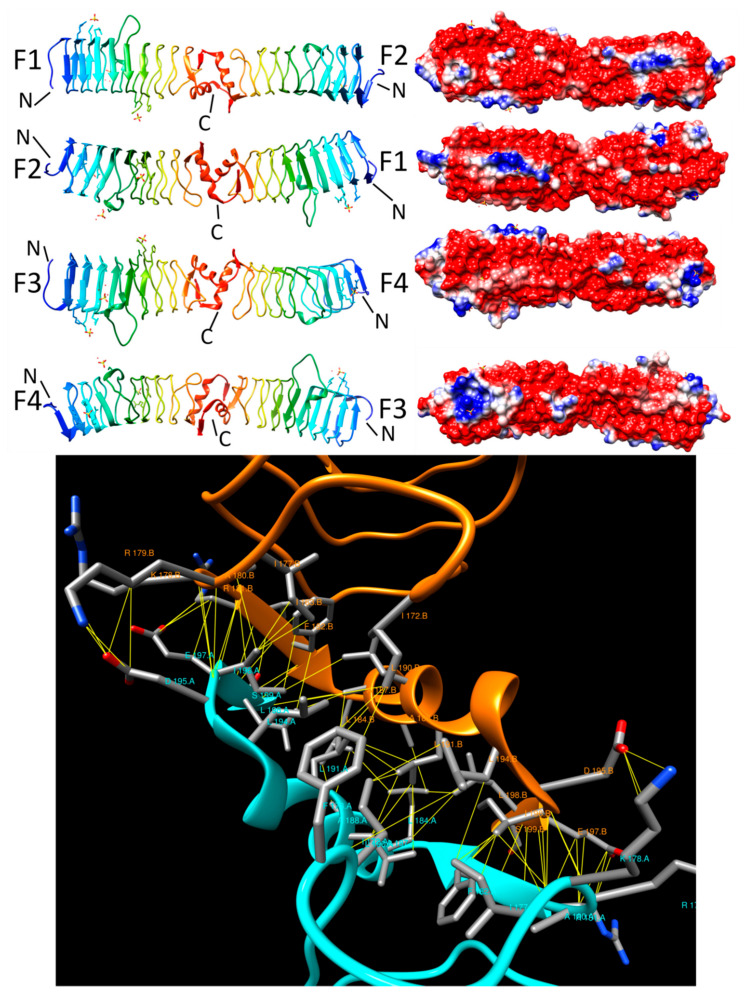
Structure and analysis of AlbG. The top panel shows the ribbon diagrams of the structure for four different orientations (at left) and the corresponding electronic potential surface plots (at right). The bottom panel is the interface model calculated by Intersurf and depicted using Chimera.

**Figure 9 biomolecules-11-00638-f009:**
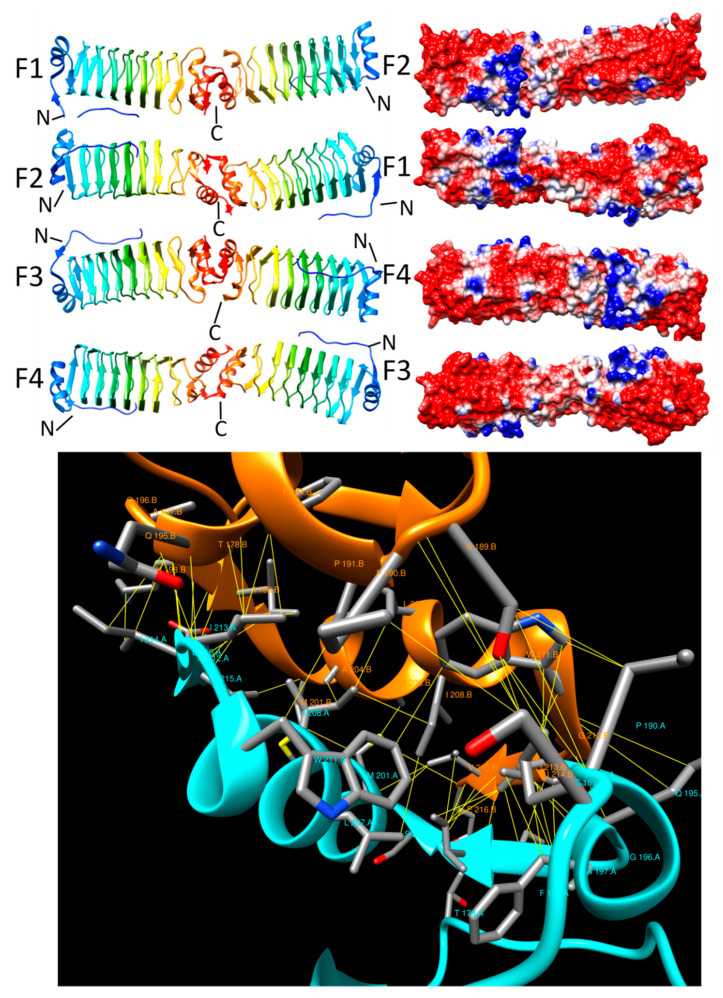
Structure and analysis of PENT. The top panel shows the ribbon diagrams of the structure for four different orientations (at left) and the corresponding electronic potential surface plots (at right). The bottom panel is the interface model calculated by Intersurf and depicted using Chimera.

**Figure 10 biomolecules-11-00638-f010:**
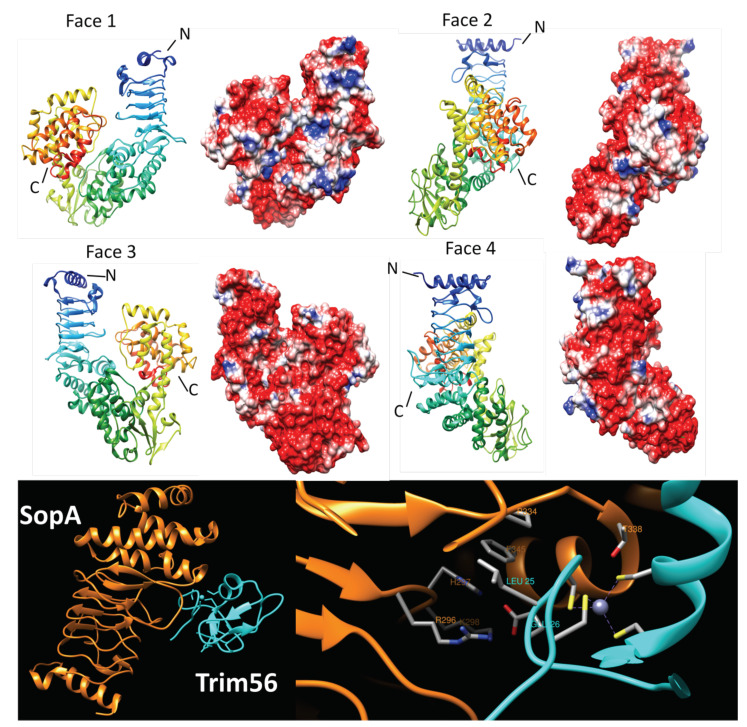
Structure and analysis of SopA. The top panel shows ribbon diagrams of the structure (PDB ID: 2QZA) for four different orientations (at left) and the corresponding electronic potential surface plots (at right). The bottom panel (left figure) shows the overall interaction between SopA and Trim56 (5JW7) and the right figure shows the details of the interaction. SopA is colored orange and Trim56 is colored cyan.

**Figure 11 biomolecules-11-00638-f011:**
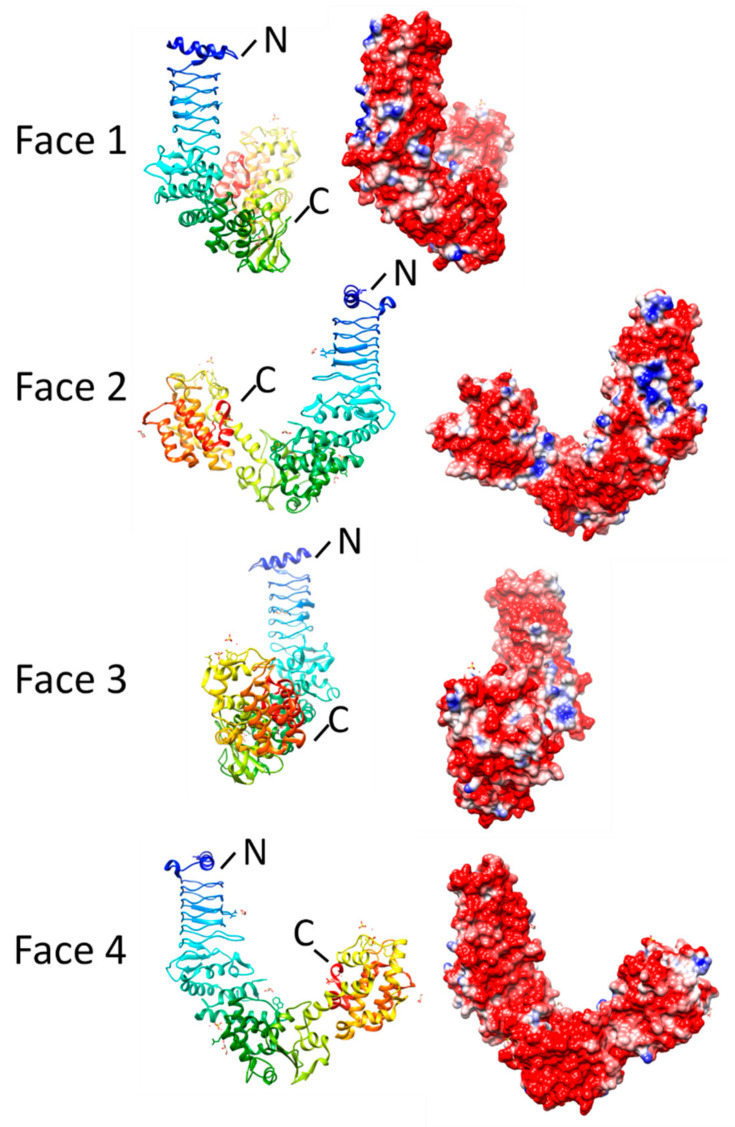
Structure and analysis of NleL. Ribbon diagrams of the structure (PDB ID: 3NAW) for four different orientations (at left) and the corresponding electronic potential surface plots (at right).

**Figure 12 biomolecules-11-00638-f012:**
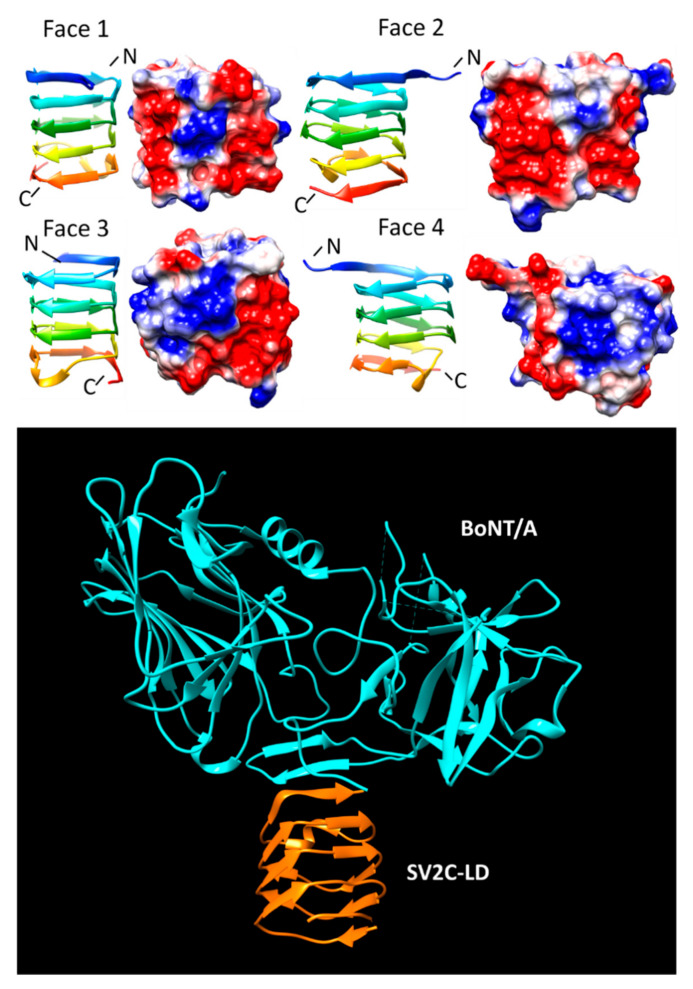
Structure and analysis of SV2C. The top panel is the ribbon diagram of the structure and electronic potential surface analysis. The bottom panel is the interface model calculated by Chimera.

**Figure 13 biomolecules-11-00638-f013:**
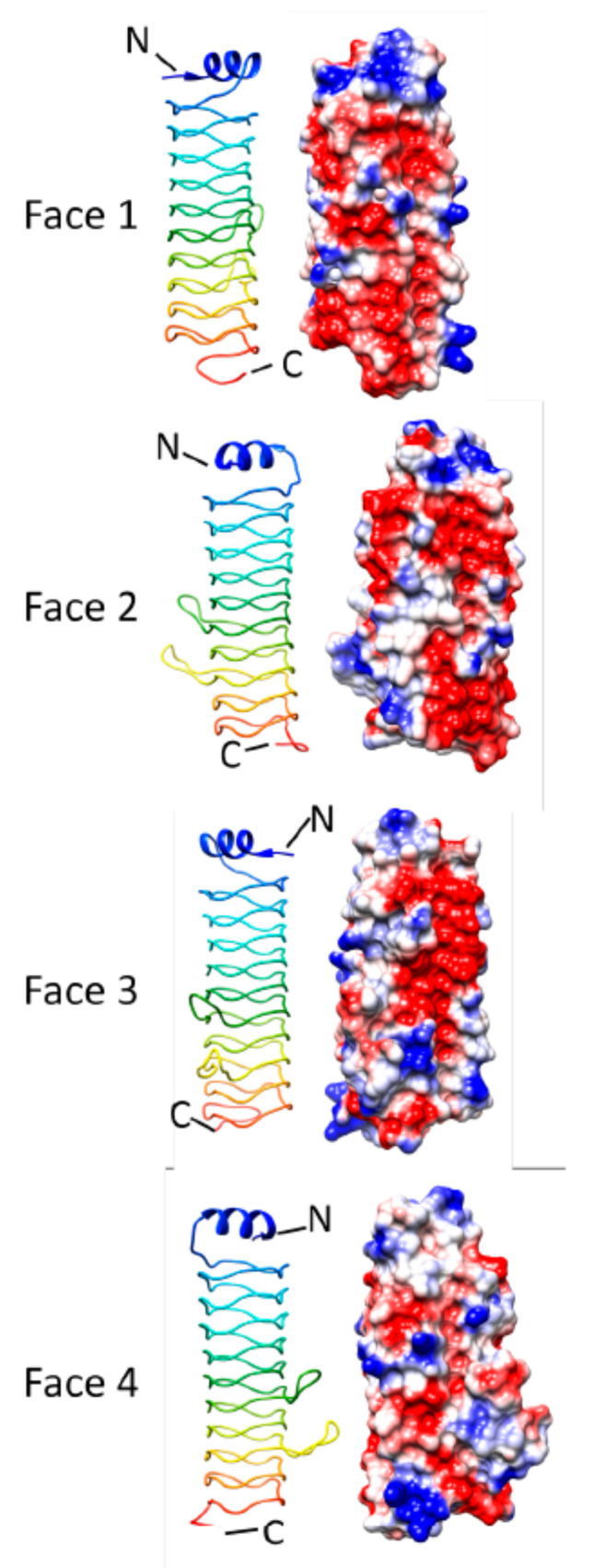
Structure and analysis of HetL. Ribbon diagrams of the structure (PDB ID: 3DU1) for four different orientations (at left) and the corresponding electronic potential surface plots (at right).

**Figure 14 biomolecules-11-00638-f014:**
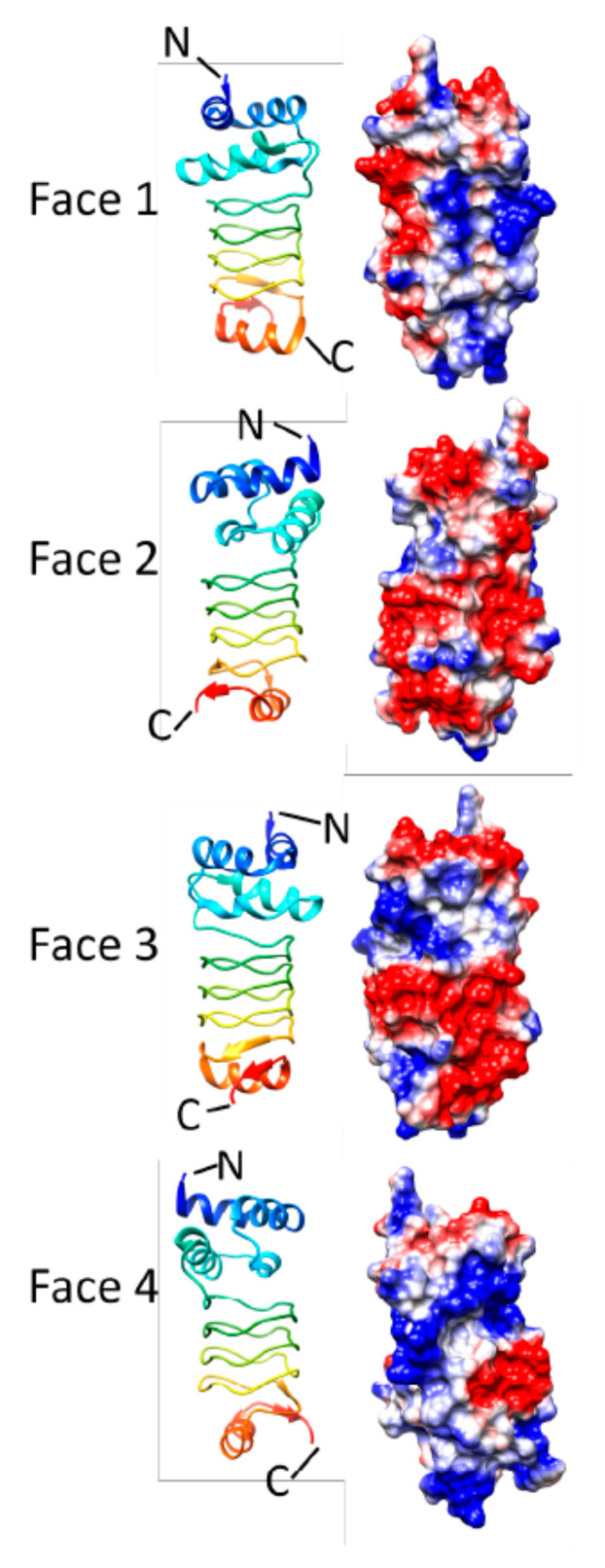
Structure and analysis of Alr1298. Ribbon diagrams of the structure (PDB ID: 6UV7) for four different.

**Figure 15 biomolecules-11-00638-f015:**
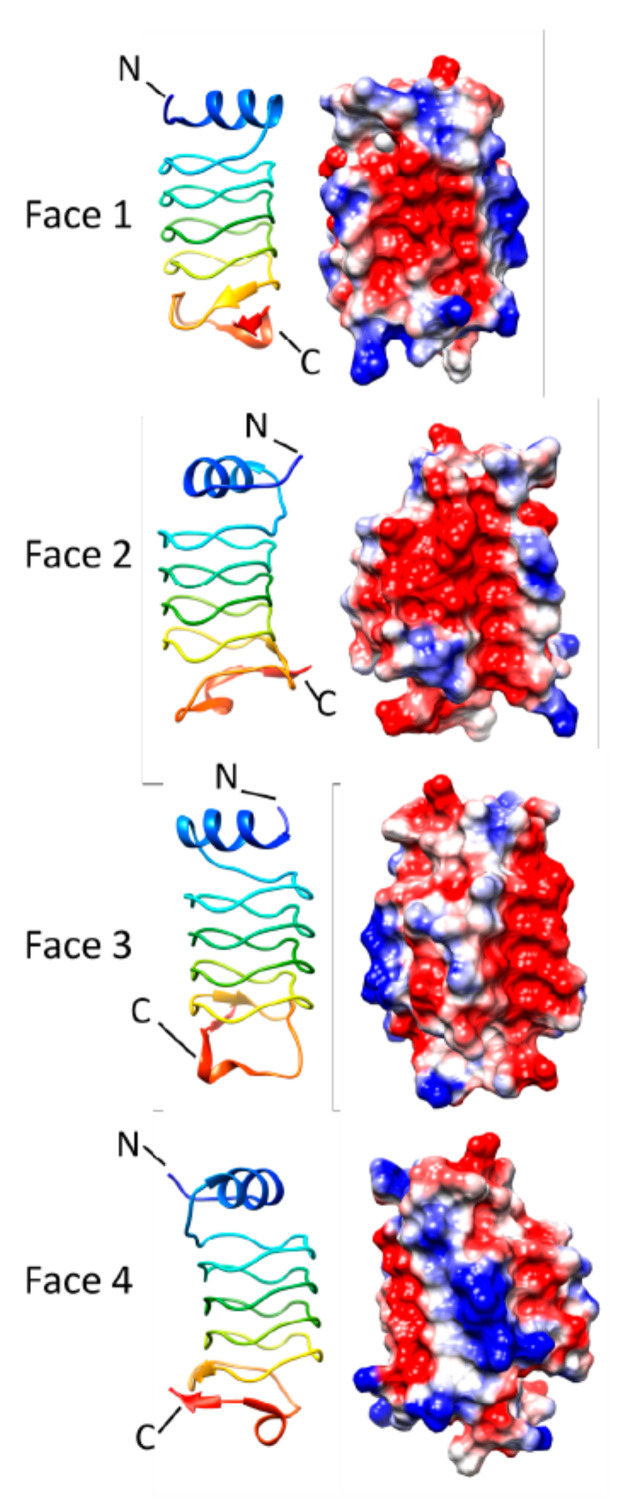
Structure and analysis of Alr5209. Ribbon diagrams of the structure (PDB ID: 6OMX) for four different orientations (at left) and the corresponding electronic potential surface plots (at right).

**Figure 16 biomolecules-11-00638-f016:**
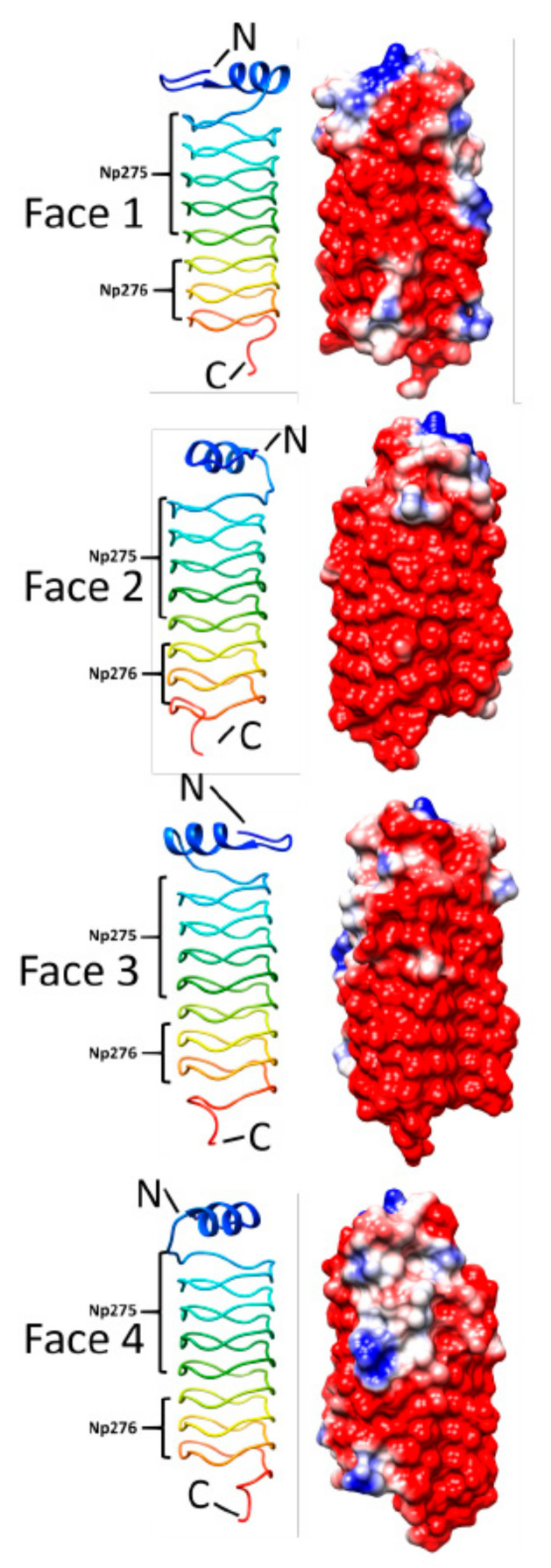
Structure and analysis of Np275/276. Ribbon diagrams of the structure (PDB ID: 2J8K) for four different.

**Figure 17 biomolecules-11-00638-f017:**
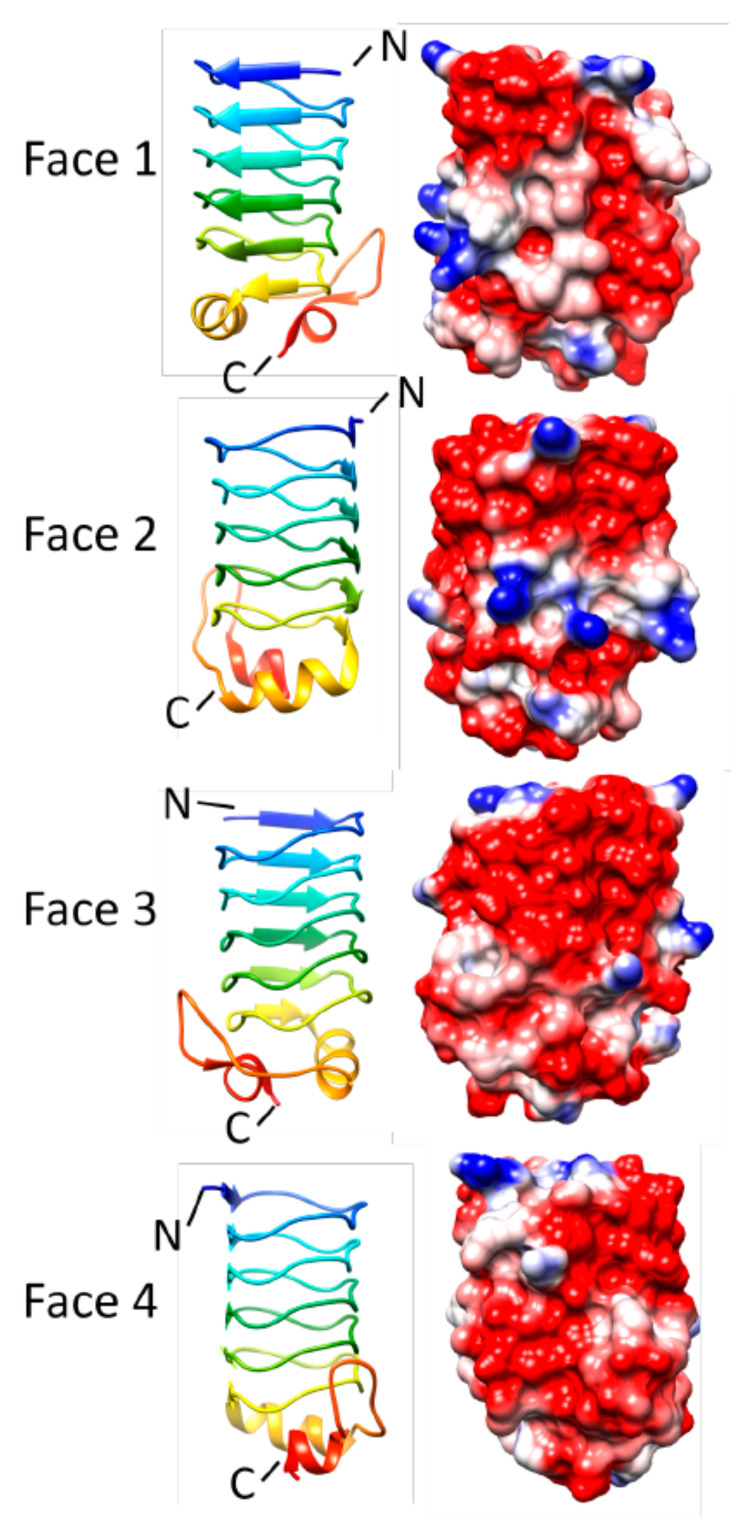
Structure and analysis of Rfr32. Ribbon diagrams of the structure (PDB ID: 2G0Y) for four different orientations (at left) and the corresponding electronic potential surface plots (at right).

**Figure 18 biomolecules-11-00638-f018:**
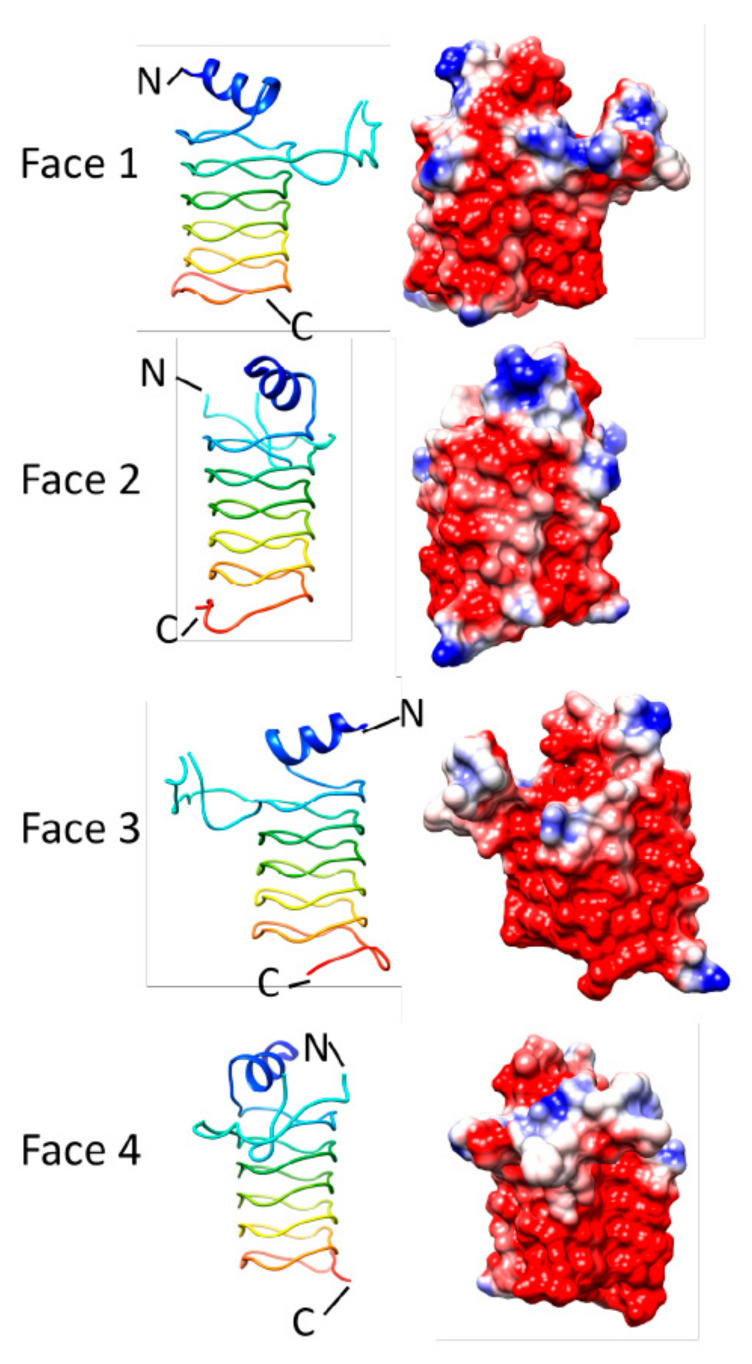
Structure and analysis of Rfr23. Ribbon diagrams of the structure (PDB ID: 2O6W) for four different.

**Figure 19 biomolecules-11-00638-f019:**
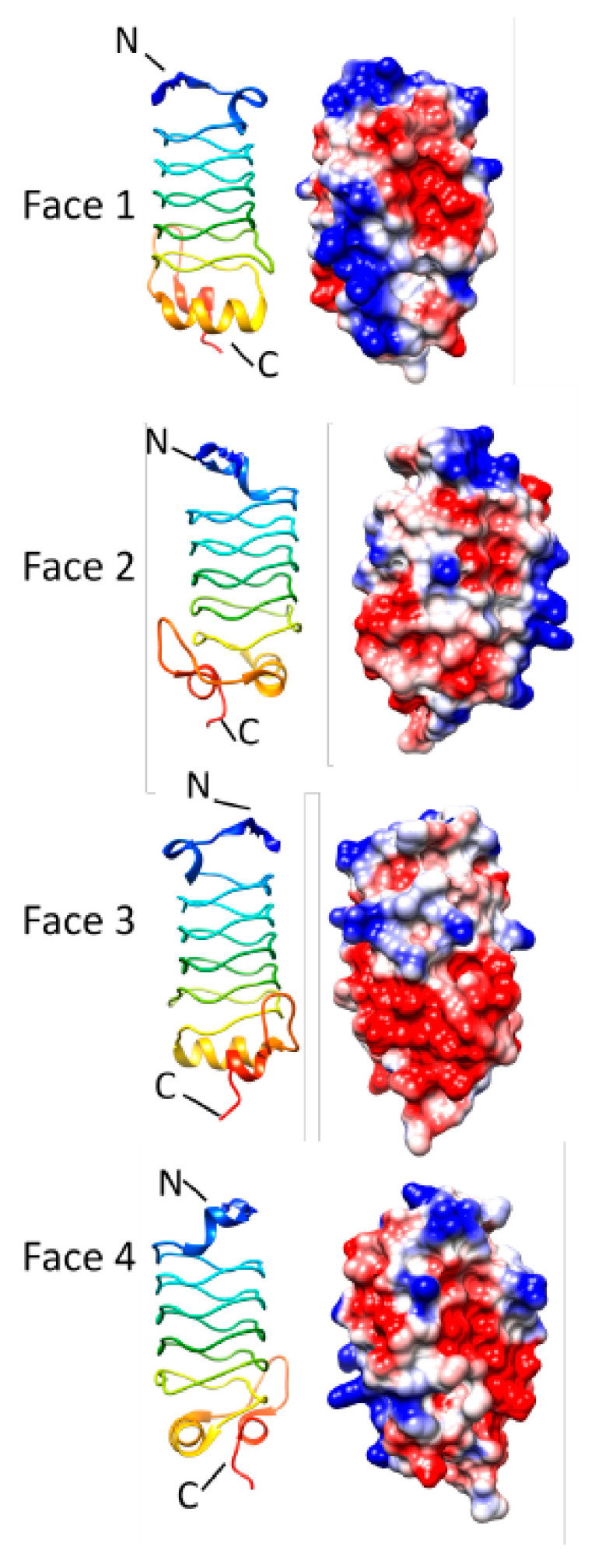
Structure and analysis of At2g49920.2. Ribbon diagrams of the structure (PDB ID: 3N90) for four different orientations (at left) and the corresponding electronic potential surface plots (at right).

**Figure 20 biomolecules-11-00638-f020:**
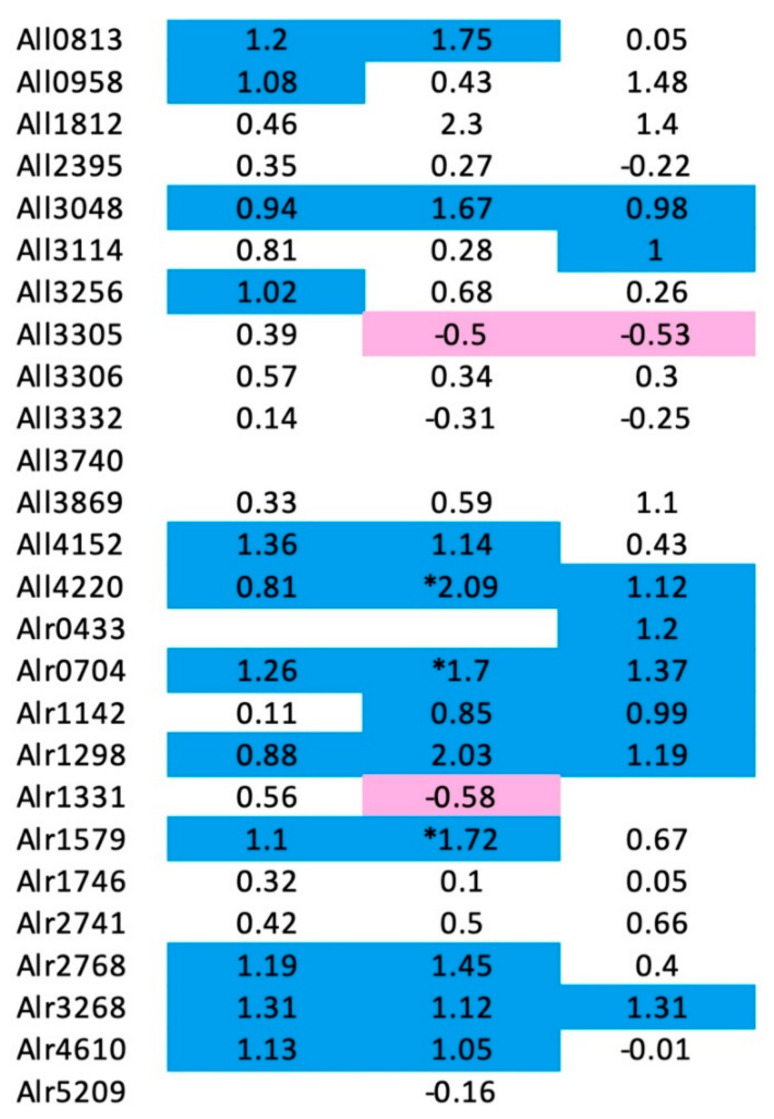
Summary of PRP gene expression in *Nostoc* sp st PCC 7120 following nitrogen deprivation. The numbers indicate the log base 2 of the relative change following nitrogen deprivation relative to the 0 h time point. Positive values indicate increased expression and negative values indicate decreased expression. Values shaded in blue indicate at least a twofold increase in expression. Values marked by an asterisk (*) indicate statistically significant changes. Values shaded in pink indicate at least a 40% decrease in expression. Values were obtained from the supplementary tables reported by Ehira and Omori [67].

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
