# Peer review of "Current Understanding of the Structure and Function of Pentapeptide Repeat Proteins"

_biomolecules, 2021, doi:10.3390/biom11050638_

Round 1
Reviewer 1 Report
This is an interesting review of a widespread but enigmatic class of proteins. It should serve as a useful stimulus for further work on their function, especially in cyanobacteria. I have only minor comments to make:
- Line 136-137. Actually Fox- means unable to fix dinitrogen except in an oxygen-depleted environment.
- line 407. It should be a small t in "typhimurium". And what is "rod-headed" supposed to mean? Surely should be "rod-shaped"?
- Line 650. Ref32 should be Rfr32!
- The references to annotations in the Uniprot database (lines 651,653, 669) are very unhelpful. I think these annotations are justified only by reference to the author's own paper (ref 13), which is the only appropriate citation for these assertions.
- Line 709. The oxygenic filamentous cyanobacteria are not an "organism", but a whole cluster of species. And it's a bit odd to single out the filamentous cyanobacteria when the PRPs are also widespread in unicellular cyanobacteria, which are equally important. Indeed, big chunks of the text are devoted to PRPs in Synechocystis and Cyanothece, which are unicellular species (lines 163-186; 642-672).
Author Response
- Line 136-137. Actually Fox- means unable to fix dinitrogen except in an oxygen-depleted environment.
comment - agreed. replaced text with "lack of ability to fix dinitrogen except in an oxygen-depleted environment"
- line 407. It should be a small t in "typhimurium". And what is "rod-headed" supposed to mean? Surely should be "rod-shaped"?
comment- correct on both counts. fixed as suggested
- Line 650. Ref32 should be Rfr32!
comment - correct! corrected.
- The references to annotations in the Uniprot database (lines 651,653, 669) are very unhelpful. I think these annotations are justified only by reference to the author's own paper (ref 13), which is the only appropriate citation for these assertions.
comment - actually, for Rfr32, in reference 13, we did not predict the sub-cellular location and did not report evidence that Rfr32 expression was observed at the protein level. This information only comes from the Uniprot database, so we feel that we have to keep this as the source of the information reported in lines 651 and 653. The same is true for Rfr23 for line 669.
- Line 709. The oxygenic filamentous cyanobacteria are not an "organism", but a whole cluster of species. And it's a bit odd to single out the filamentous cyanobacteria when the PRPs are also widespread in unicellular cyanobacteria, which are equally important. Indeed, big chunks of the text are devoted to PRPs in Synechocystis and Cyanothece, which are unicellular species (lines 163-186; 642-672).
comment - agreed, that was a mistake. We have removed "filamentous" from that statement.
Reviewer 2 Report
This review on PRP by Zhang and Kennedy is extensive, well organized, and highly detailed. PRPs are highly worthy of study as they adopt unusual structures that are involved in diverse functions, but in ways that remain mysterious. Thus, this review will serve as a useful resource for both newcomers and established investigators in this field, which has much room to grow as our understanding of this superfamily is woefully incomplete.
Author Response
No corrections required. Thank you.